

# Full characterization and calibration of a transfer standard monitor for atmospheric radon and thoron measurements

Roger Curcoll[1], Claudia Grossi[1,2], Stefan Röttger[3], Arturo Vargas[1]

[1]Institut de Tècniques Energètiques (INTE), Universitat Politècnica de Catalunya, Barcelona, Spain
5  [2]Department of Physics, Universitat Politècnica de Catalunya, Barcelona, Spain
[3]Physikalisch-Technische Bundesanstalt, 38116 Braunschweig, Germany

*Correspondence to*: Roger Curcoll (roger.curcoll@upc.edu)

**Abstract.**

10  In this work a full characterization of the new user-friendly version of the Atmospheric Radon MONitor (ARMON), used to measure very low activity concentrations of the radioactive radon gas in the outdoor atmosphere, is carried out. The ARMON is based on the electrostatic collection of $^{218}Po^+$ particles on a semiconductor detector surface. A main advantage of this instrument is offering high resolution alpha energy spectra which will allow to separate radon progeny ($^{210}Po$, $^{218}Po$ and $^{214}Po$). The monitor feature may also allow measurements of thoron ($^{220}Rn$) by collection of $^{216}Po^+$.

In this work the physical principle, the hardware configuration and the software development of the automatic and remotely controlled ARMON, conceived and constructed within the MAR$^2$EA and the traceRadon projects, are described. The monitor efficiency and its linearity over a wide spam of radon concentration activities has been here evaluated and tested using theoretical as well as experimental approaches. Finally, a complete budget analysis of the total uncertainty of the monitor was 20  also achieved.

Results from the application of a simplified theoretical approach shows a detection efficiency for $^{218}Po^+$ of about 0.0075 (Bq m$^{-3}$)$^{-1}$ s$^{-1}$. The experimental approach, consisting of exposing the ARMON at controlled radon concentrations between few hundreds to few thousands of Bq m$^{-3}$, gives a detection efficiency for $^{218}Po^+$ of 0.0057 ± 0.0002 (Bq m$^{-3}$) s$^{-1}$. This 25  last value and its independence from the radon levels was also confirmed thanks to a new calibration method which allows, using low emanation sources, to obtain controlled radon levels of few tens of Bq m$^{-3}$.

The total uncertainty of the ARMON detection efficiency obtained for hourly radon concentration above 5 Bq m$^{-3}$ was lower than 10 % (*k*=1). The characteristics limits of the ARMON were also calculated, being those dependent on the presence of 30  thoron in the sampled air, and a value of 0.132 Bq m$^{-3}$ was estimated in thoron absence. Current results may allow to confirm that the ARMON is suitable to measure low-level radon activity concentration (1 Bq m$^{-3}$ - 100 Bq m$^{-3}$) and to be used as transfer standard to calibrate secondary atmospheric radon monitors.

## 1 Introduction

$^{222}Rn$ is a radioactive noble gas naturally generated from Radium ($^{226}Ra$) within the primordial Uranium-238 ($^{238}U$) decay 35  chain (Nazaroff and Nero, 1988). Its exhalation from soils depends mainly on the uranium content, soil properties as porosity or bulk density, and soil moisture (Conen and Robertson, 2002). The global $^{222}Rn$ source into the atmosphere is mainly restricted to land surfaces (Szegvary et al., 2009; Karstens et al., 2015), with the $^{222}Rn$ flux from water surfaces considered negligible for most applications (Schery and Huang, 2004). Radon has a half-life of 3.82 days and, due to the fact that does not have any other significant atmospheric sink rather than its radioactive decay, has been largely used in the last decades as a



tracer for atmospheric studies. $^{222}$Rn has been used to understand atmospheric processes such as the dynamics of the boundary layer (Chambers et al., 2011; Pal et al., 2015; Vargas et al., 2015), to improve inverse transport models (Hirao et al., 2010), to assess the accuracy of chemical transport models ( Jacob and Prather, 1990; Arnold et al., 2010; Chambers et al., 2019), or to study atmospheric transport and mixing processes within the planetary boundary layer ( Zahorowski et al., 2004; Galmarini, 2006; Baskaran, 2011, 2016; Williams et al., 2016). When measured together with another gas (e.g. air pollutants or greenhouse

gases such as carbon dioxide or methane), it can be also used to detect sources and to indirectly quantify fluxes of that gas. The Radon Tracer Method (RTM) (Levin et al., 1999) is one of the methodologies used to indirectly determine regional and nocturnal fluxes of greenhouse gases and air pollutants (Vogel et al., 2012; Wada et al., 2013; Levin et al., 2021). In addition, if RTM is used together with back trajectories analyses, it will allow a better quantification of the different local versus regional contributions and an estimation of the effective radon flux seen by the station under study (Grossi et al., 2018).

For its utility, $^{222}$Rn measurements are so far not mandatory but recommended at the atmospheric stations of the Integrated Carbon Observation System network (ICOS RI, 2020). Atmospheric radon activity concentrations are usually ranging between few hundreds of mBq m$^{-3}$ and tens of Bq m$^{-3}$, depending if the measurements are carried on at coastal or continental sites, respectively (Chambers et al., 2016; Grossi et al., 2016). Thus, high precision radon measurements are required for atmospheric applications.

Available commercial radon monitors, usually used in the radiation protection field or for geophysics research goals, are so far not suitable for high quality atmospheric measurements too (Radulescu et al., 2022). In the last years three research entities have designed and developed high sensitive $^{222}$Rn or $^{222}$Rn progeny monitors which are currently running at different European atmospheric stations: i) the Heidelberg monitor, developed at the Institute of Environmental Physics of the Heidelberg University (Schmidt et al., 1996; Levin et al., 2002), determines the atmospheric $^{222}$Rn activity concentration using the

measured $^{214}$Po daughter activity with a static filter method and assuming a constant equilibrium factor between radon and its short lived progeny in air. (Schmithüsen et al., 2017); ii) the monitor from the Australian Nuclear Science and Technology Organisation (ANSTO), which determines the atmospheric $^{222}$Rn activity concentration using dual-flow-loop two-filter method (Whittlestone and Zahorowski, 1998; Zahorowski et al., 2004); iii) the Atmospheric Radon MONitor (ARMON), designed and built at the Institute of Energy Technologies (INTE) of the Universitat Politècnica de Catalunya (UPC), which is based on

alpha spectrometry from the positive ions of $^{218}$Po electrostatically collected on a Passivated Implanted Planar Silicon (PIPS) detector surface (Grossi et al., 2012; Vargas et al., 2004, 2015). Several monitors of this last type have been displaced at atmospheric Spanish stations for atmospheric research studies (Grossi, 2012; Hernández-Ceballos et al., 2015; Vargas et al., 2015; Grossi et al., 2018; Gutiérrez-Álvarez et al., 2019). The response of the ARMON under different field conditions was also compared with the ones from other previously cited research instruments in the south of Paris in 2016 (Grossi et al., 2020).

In the framework of the Catalan MAR$^2$EA project (High Efficiency monitor of atmospheric radon concentration for radioprotection and environmental applications, Llavor program, 2020-2021) and of the Working Package 1 (WP1) of the European traceRadon project (Röttger et al., 2021), an improved ARMON prototype was developed (here labelled as ARMON v2). The main objective of the traceRadon project was the development of a metrological infrastructure to ensure traceable of low levels radon measurements. Specifically, the WP1 aimed to develop traceable methods, according to IEC 61577, for the

measurement of outdoor low-level radon activity concentrations in the range of 1 Bq m$^{-3}$ to 100 Bq m$^{-3}$, with uncertainties lower than 10 % ($k$=1), to be used in climate and radiation protection networks. Within this WP1, the INTE-UPC group was in charge to design and build a mobile and user-friendly transfer standard instrument useful to calibrate radon monitors running at atmospheric and radiological stations. This new user-friendly monitor is an improved version of the previous ARMON, mainly in regard to its robustness, portability, sensitivity, setting and automatic control.




In the present manuscript the design and setup of the ARMON v2 are described in detail together with the theoretical and experimental methodologies applied to evaluate the detection efficiency of the monitor. The total uncertainty of the ARMON detection efficiency was also calculated considering the different parameters and variables that could influence it such as the statistic number of counts of each alpha spectrum measured by the ARMON v2, the effect of the water content of the sampled

air, the STP (Standard Temperature and Pressure) correction, the monitor background, etc.

The ARMON was calibrated at the INTE-UPC radon chamber using reference radon concentrations between few hundred Bq m$^{-3}$ and few thousand Bq m$^{-3}$. In order to check if the detection efficiency obtained thanks to the INTE-UPC exposures was also confirmed for very low radon activities concentrations (tens of Bq m$^{-3}$), an independent experiment was carried out at the

Physikalisch-Technische Bundesanstalt (PTB) facility, using new low radon emanation sources and methods also made generated within the WP1 of the traceRadon project.
The ARMON v2 presented in this paper, thanks to its sensitivity and robustness, has the potential to help in the improvement of the sources, transport, and fate of $^{222}$Rn in the environment. The full characterization of this instrument and its uncertainty budget may be useful to support the development of accurate atmospheric studies and to enhance the capabilities of the Radon

Tracer Method for estimating GHG fluxes.

## 2 ARMON description

### 2.1 Physical principles of the ARMON v2

The physical principle of operation of the ARMON is based on the collection of the positive $^{218}$Po charged particles, due to the alpha decay of the $^{222}$Rn within the detection volume, on the surface of a semiconductor detector. This methodology is well

known and has been used in the past by other researchers (Hopke, 1989; Tositti et al., 2002; Grossi et al., 2012). $^{218}$Po$^+$ particles, generated within a known volume, are found to be in the form of singly charged positive ions the 88 % of the time, while the neutral ions occur the remaining 12 % of the time (Goldstein and Hopke, 1985). $^{218}$Po$^+$ can be due to the stripping of orbital electrons by the departing α particle or by the recoil motion. When a high electric potential is applied to the internal surface of the detection volume and the detector itself is maintained at 0 V, an Electrostatic Field (EF) is generated inside the volume,

making the charged $^{218}$Po$^+$ particles to be collected at the detector surface within short time.
In the case of the ARMON, a Passivated Implanted Planar Silicon (PIPS) detector is used. A preamplifier and an amplifier are then used to amplify and shape the electric signal coming from the detector to a Gaussian function in order to be read by a multichannel analyser (MCA), that transforms it into counts for specific energy bins. The spectra generated are then analysed with the software MAESTRO (Multichannel Analyzer Emulation Software, ORTEC). A typical one-hour spectrum from

atmospheric radon in air obtained with the ARMON v2 is shown in the Appendix A (Fig. A1).
Using this previous methodology, the $^{218}$Po counts (with an α decay at 6.0 MeV) can be separated in the spectrum from other $^{222}$Rn progeny isotopes such as the $^{214}$Po (α decay at 7.7 MeV) and the $^{210}$Po (α decay at 5.3 MeV). Using the same principle, the ARMON v2 is also able to measure $^{220}$Rn by detection of its progeny $^{216}$Po (α decay at 6.8 MeV) and $^{212}$Po (8.78 MeV). However, in the present manuscript the full characterization of the instrument was only carried out for radon measurements

due to the lack of metrology chain for low-level thoron measurements. In this regard, it is needed to be clarified that if $^{220}$Rn (thoron) is also present within the sampled air, $^{212}$Bi particles, due to its decay chain, are also formed through β-decay of $^{212}$Pb. The 36 % of this $^{212}$Bi α-decays to $^{208}$Tl at a similar energy than $^{218}$Po (6.05 MeV) and affects the net counts of $^{218}$Po and thus the uncertainty of the final radon measurements, as explained in Grossi et al., 2012 and Vargas et al., 2015. The other 64 % of the $^{212}$Bi particles β-decay to $^{212}$Po (t$_{1/2}$ = 3.0 10$^{-7}$s), which α-decays at 8.78 MeV to the stable nuclide $^{208}$Pb. Thanks to the

high energy resolution of the ARMON spectra, the decay of the $^{212}$Po particles can be registered, separated and counted. Therefore, the $^{212}$Bi counts can be estimated by multiplying the factor 36/64 to the $^{212}$Po counts, and its contribution may be



subtracted from the gross $^{218}$Po counting. The radon concentration is thus calculated for each spectrum of real time length $t$ (in seconds), from the Eq. (1):

$$C_{Rn} = \left[ \frac{nc_{Po218}}{t} - \left( \frac{nc_{Po212}}{t} \frac{36}{64} \right) \right] \frac{1}{\varepsilon} \quad (1)$$

Where $nc_{Po218}$ is the number of counts detected within the ROI (region of interest) of $^{218}$Po, $nc_{Po212}$ is the number of counts detected within the ROI of $^{212}$Po, $t$ is the integration time of the spectrum, $\varepsilon$ is the detection efficiency of the instrument, defined as detected $^{218}$Po count rate per $^{222}$Rn air concentration, and here expressed in counts per seconds (cps, s$^{-1}$) per Bq m$^{-3}$. It is also important to underline that charged $^{218}$Po ions present within the detection volume may be neutralized due to the interaction with water vapour present in the sampled air via the formation of hydroxyl radicals OH (Hopke, 1989). Therefore,

water vapour particles must be kept as low as possible inside the detection volume in order to maximize the collection efficiency and the response of the monitor, under different water vapour content conditions, must be corrected as shown here. Assuming a linear correction of the efficiency due to water vapour concentration (Hopke, 1989), the real efficiency of our monitor can be expressed by Eq. (2), where $\varepsilon_0$ is the efficiency in dry condition (0 ppmv of H$_2$O), $b$ is the trend of the linear correction and $[H_2O]$ is the water vapour concentration.

$$\varepsilon = \varepsilon_0 - b[H_2O] \qquad (2)$$

## 2.2 ARMON set up: Hardware and Software

A schematic design of the ARMON v2 is shown in Fig. 1a. A photography of the external case is showed in Fig. 1b. Before entering the detection volume, the air, sampled with a pump, (blue line in Fig. 1a) passes through a 0.5 µm filter to prevent the entry of dust and aerosol attached $^{222}$Rn progeny into the detection volume. Then the air enters into the detection volume which

is made by a glass sphere inside silver-plated with a neck of 45 mm of inner diameter. The PIPS detector of 300 mm$^2$ active area (Mirion Technologies A300-17) is located on the upper part of the sphere, tangent to it and at the bottom of the neck. This last configuration was used to maximize the collection of the polonium by the EF as shown later and it was obtained using a solid Teflon stopper. A high voltage power supply (Glassman MJ15P1000) provides a potential of 10 kV between the PIPS detector (at 0 V) and the sphere walls to create the EF. The pulse pre-amplifier and amplifier (model: CR-10 from Pyramid

Technical Consultants Inc.) is located outside the sphere to shape and to amplify the signal and to send it to the MCA (model: ORTEC EASY-MCA 2k). When the sampled air exits the detection volume, it passes through a series of sensors: a digital flow meter (SMC PFM710S-F019), a temperature meter (JUMO PT100) and a dew point meter (VAISALA DMT 143). The sensors are controlled by a datalogger (Advantech USB-4622-CE) connected to a laptop. All the hardware is installed inside a flight-case box of 128x50x50 cm$^3$, with the inlet and outlet air sampling connectors located on the backside of the case (Fig

1b). The different components of the instrument are placed on different trays and drawers in order to easily access to them and make the necessary maintenance if needed. A drawing and photos of the monitor are shown in the Appendix A (Fig. A2). The inlet flow required for the monitor is of about 2 L min$^{-1}$ of dried air.





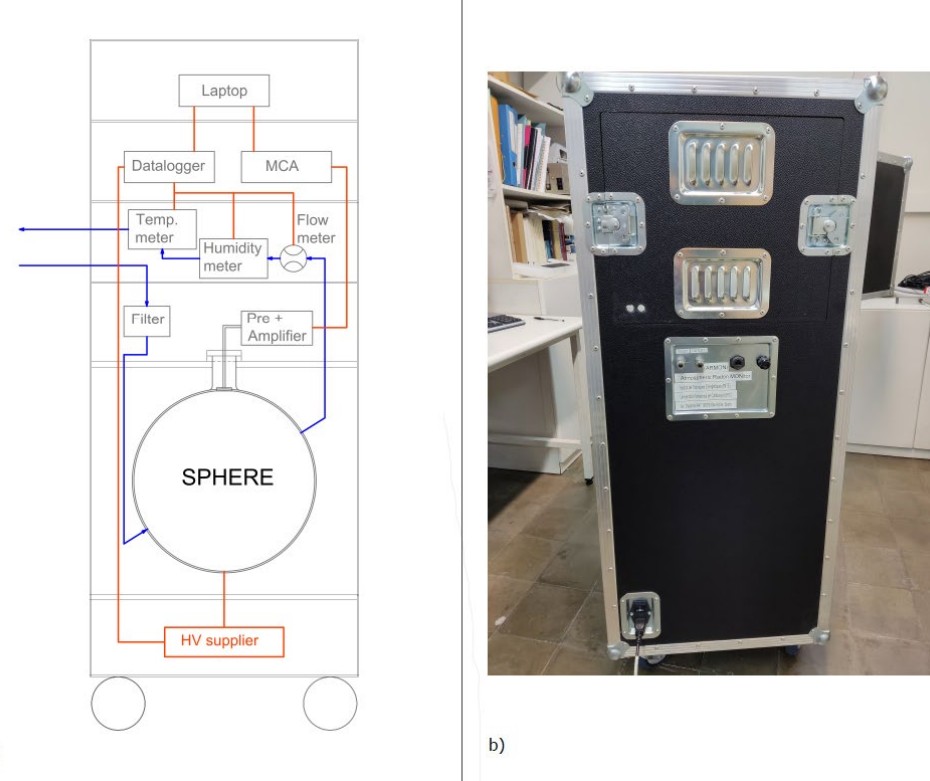

a)                                                                      b)

**Figure 1. a): schematic design of the ARMON v2 with its main hardware's and their location. b): Image of the backside of the**
**instrument.**

A specific software named ARMON_LAB, built on LabVIEW® (Laboratory Virtual Instrument Engineering Workbench), was
developed in order to monitor and to control all the parameters and variables of the instrument with the help of the Advantech®
datalogger. The software is installed in the ARMON v2 laptop to give the user a full control of the monitor. The software
allows the visualization in Real Time of the different sensors' outputs (flow, humidity and temperature) and allows the control
of the high voltage applied to the detection volume. Ten minutes' averages of the different variables are automatically saved
in daily files. In parallel, the spectra obtained by the MCA are automatically and regularly saved using the Maestro software
script (ORTEC, 2012). After each measurement (usually working on hourly base for atmospheric stations requirements but it
can be easily modified) the ARMON_LAB software calls an R script which uses the information from the Maestro and the
output from the environmental sensors to calculate the radon concentration. Real-time as well as past radon concentrations
data can be visualized within the ARMON_LAB interface. Two screenshots of the software are shown in the Appendix A (Fig.
A3). The laptop can be connected to internet whether by Wi-Fi or by an ethernet wire and, once installed, the instrument can
be fully remotely controlled. A flow chart of the data of the ARMON v2 monitor is shown in the Appendix A (Fig. A4).




**2.3 ARMON v2 detection efficiency**

**2. 3.1 Theoretical approach**

In order to calculate the radon concentration measured by the ARMON v2 with Eq. (1), the total efficiency ($\varepsilon$) of the instrument
needs to be known with the possible lowest uncertainty. First of all, the order of magnitude of this efficiency was evaluated
using a simplified theoretical approach. The theoretical detection efficiency of the ARMON v2, $\varepsilon_t$, can mainly be factored in
two terms: the geometric contribution ($\varepsilon_g$) due to the geometry of the detector surface and corona and the collection efficiency
($\varepsilon_c$) that depends on the efficiency of the collection of the $^{218}$Po$^+$ on the detector active surface. The two contribution are

expressed in Eq. 3:

$\varepsilon_t = \varepsilon_g \cdot \varepsilon_c$  (3)

The analysis of these two factors allowed to optimize them during the building of the monitor. As commented in section 2.1,
the maximum possible percentage of positive charged $^{218}$Po ions collected over the detector surface is of 88 % (Hopke, 1989)
($p_{218_{Po^+}}$). However, both the active surface of the PIPS detector and the not active surface (the corona) are at the same potential

(0 V), so when the ions reach the detector they will be distributed over the entire surface, both on the active part and on the
non-active one. Luckily, this distribution is not spatially homogenous and it will depend on the symmetry and geometry of the
generated EF as it will be shown here. Furthermore, of those particles collected at the active surface ($p_{Active}$), only about the
50 % will be emitting alpha particles on the plane including the detector and therefore counted ($p_{Detected}$). The number of ions
per second that are formed in the sphere for a radon concentration in air of 1 Bq m$^{-3}$ are calculated by multiplying the formed

ions $p_{218_{Po^+}}$ by the sampled air volume $V$ (0.02 m$^3$) and then multiplied by the percent of ions arriving on the detector surface
and emitting in the detector plane. The resulting $\varepsilon_g$ in s$^{-1}$ per Bq$^1$ m$^{-3}$ is calculated according Eq. (4).

$\varepsilon_g = V \, p_{218_{Po^+}} \cdot \, p_{Active} \cdot p_{Detected}$          (4)

In order to understand and thus to maximize the collection of the polonium ions on the detector surface, the software COMSOL
Multiphysics was used to simulate the shape of the EF generated within the ARMON v2 detection volume when different kV

of electric potential ($V$) were applied to the sphere wall. The COMSOL is based on the solving of equations for finite
element analysis. The output of the COMSOL simulation, with the value of the simulated electrostatic field at each spatial grid
of the ARMON v2 detection volume, was then used to calculate the drift velocity, the collection trajectories and the travelling
time of 10.000 polonium fictitious particles, which were initially randomly spaced within the volume. The instantaneous drift
velocity for each particle $i$ inside the detection volume depends on the mobility ($\mu$) of the $^{218}$Po$^+$ particles and the EF at its

position as reported in Eq. 5:

$v_i = \mu \vec{E_i}$          (5)

The mobility of the $^{218}$Po$^+$ ions in air is known to be between 1 cm$^2$ (V s)$^{-1}$ and 6 cm$^2$ (V s)$^{-1}$ (Nazaroff and Nero, 1988;
Pugliese et al., 2000).  A mobility of 3 cm$^2$ (V s)$^{-1}$ was recently reported by Symour (2017) for a similar study and was also
used in the present study. Trajectories were calculated using time steps of 10 µs. The arriving position of the simulated particles

on the detector surface were used to estimate the percentage of polonium particles collected on the active area (2.99 cm$^2$) and
on the not active area (3.44 cm$^2$) of the detector.

The percentage of polonium ions arriving on the detector surface was calculated taking into account if during the travelling
time they will go under radioactive decay ($T_{1/2}$ = 184.3 s.) or if they will be neutralized due to their recombination with OH$^-$
particles or small positive air ions (Dankelmann et al., 2001). At this regard, Hopke (1989) found that this recombination

depends on the water volume concentration and that the interval time $\tau_{H_2O}$, for $^{218}$Po recombination in an electrostatic chamber



had a value of 0.879 [H2O]$^{-1/2}$. From the calculated travelling time, equal to the ratio between the trajectory of each particle to reach the detector and its drift velocity, the effect of the recombination with water particles was calculated as Eq. (6):

$$N = N_o e^{-\log(2)\, t/(0.879\,[\text{H2O}]^{-1/2})} \quad (6)$$

where $N$ are the particles that has not been recombined within the travelling time t, $N_o$ is the initial number of particles and

0.879 [H2O]$^{-1/2}$ is the interval time of recombination with $H_2O$ for $^{218}$Po$^+$ particles. Finally, the theoretical collection efficiency $\varepsilon_c$ will be calculated as $N/N_o$.

The theoretical efficiency $\varepsilon_t$ obtained from Eq. 3 is has been calculated under ideal conditions hypothesis. However, the real geometry of the generated EF could not be so regular due to: i) the difficulty of positioning the PIPS surface tangent to the

sphere; ii) inhomogeneity present in the layer of the cover conductive material of the internal wall of the sphere; iii) uncertainty in the determination of the potential $V$ applied to the sphere, and iv) the spherical shape and exact measure of the detection volume. Thus, the real efficiency of the monitor could be lower than $\varepsilon_t$ and it needs also to be evaluated experimentally.

### 2.3.2 Experimental approach

The experimental detection efficiency of the ARMON v2 was obtained by comparing the detected net counts of $^{218}$Po measured

with the instrument with a reference radon activity concentration $C_{Ref}$ measured with a secondary standard reference instrument as it will be explained in the following lines.

The ARMON v2 was calibrated at the INTE-UPC STAR (System for Test Atmospheres with Radon) (Vargas et al., 2004) in October 2021. The INTE-UPC STAR is a chamber with a volume of 20 m$^3$ which allows to set-up and to continuously measure

the radon activity concentration (range 200 Bq m$^{-3}$ to 30 kBq m$^{-3}$), the temperature (range 10 ºC - 40 ºC) and the relative humidity (range 15 %-95 %) (Vargas et al., 2004). The radon source inside the chamber consists of an enclosed Pylon Electronics containing 2100 kBq of $^{226}$Ra. Stable radon concentration inside the chamber are reached by controlling the air flow through the enclosed source and the ventilation rate of the chamber. The second standard reference instrument of this facility is an Atmos monitor (Radonova), serial number 220030. The traceability of the measured magnitude in Bq m$^{-3}$ is

referred to the Swedish Radiation Safety Authority (Calibration certificate n. SSM2021-2989-4) with an expanded uncertainty ($k$=2) of 6.7 % for 1500 Bq m$^{-3}$.

During the experiments, the ARMON v2 detection efficiency was estimated in a range of radon concentrations between 0.5 kBq m$^{-3}$ and 6.2 kBq m$^{-3}$. The ARMON v2 and the reference monitor were installed outside the STAR in parallel configuration. For each instrument, air coming from the radon chamber was passing through monitor and then returned to the

chamber. A silica gel dryer was installed before the air was entering at the ARMON v2 in order to reduce the water concentration of the sampled air. The integration time of the ARMON v2 spectra was chosen to be 1 h, and hourly means from the ATMOS were selected from the 10 min. default integration time. Calibration experiments lasted three weeks. The average $H_2O$ concentration inside the ARMON's detection volume during the efficiency experiments was of about 300 ppmv. The influence of the water vapour concentration on the efficiency was also evaluated at different radon concentrations within the

range (635 – 5900) Bq m$^{-3}$ and within the range (100 – 3000) ppmv $H_2O$, by using different levels of saturated silica gel as dryer.

### 2.4 Uncertainty analysis and characteristic limits of the ARMON v2

The radon activity concentration with the ARMON v2 is calculated, for each acquired spectrum, from the Eq. (1) and its unit is in Bq m$^{-3}$. In order to have comparable results with radon values from other stations or monitors, the concentration can be

multiplied by a Standard Temperature and Pressure (STP) factor to standardize the concentration obtained to a referenced value of Temperature and Pressure of air. The STP factor can be calculated by Eq. (7):





$$STP = C_T \, C_P = \frac{P_{ref}}{P} \frac{T}{T_{ref}} \quad (7)$$

Where $C_T$ and $C_P$ are the corrections for Temperature and Pressure respectively, with $T$ and $T_{ref}$ are the sampling temperature and the reference temperature respectively (in K), $P$ and $P_{ref}$ are the sampling pressure and the reference pressure respectively.

Therefore, Eq. (1) can be expanded, taking into account both the corrected value of the monitor detection efficiency under different humidity conditions as expressed in Eq. (2) and the STP correction from Eq. 6 in the following Eq. (8):

$$C_{Rn} = \left[ \frac{nc_{Po218}}{t} - \left( \frac{nc_{Po212}}{t} \frac{36}{64} \right) \right] \frac{1}{\varepsilon_0 - b \, [H_2O]} \frac{P_{ref}}{P} \frac{T}{T_{ref}} \quad (8)$$

The uncertainty for the radon concentration measurement will calculated, in agreement with according Guide to the expression of uncertainty in the measurement (BIPM et al., 2008) as in Eq. (9):

$$u_{C_{Rn}}^2 = \sum_{i=1}^{n} \left( \frac{\partial C_{Rn}}{\partial x_i} \right)^2 u_{x_i}^2 \quad (9)$$

where $x_i$ are the different variables from Eq. 8 taken in consideration for the propagation of the uncertainty.

Resolving the partial differential equation and using Eq. (8), the resulting equation is given in Eq. (10):

$$u_c{}^2(C_{Rn}) = \left( \frac{C_p \, C_T}{t \, \varepsilon} \right)^2 \left( u_{nc_{Po218}} \right)^2 + \left( -\frac{C_p \, C_T}{t \, \varepsilon} \frac{36}{64} \right)^2 \left( u_{nc_{Po212}} \right)^2 + \left( -\left[ \frac{nc_{Po218}}{t} - \left( \frac{nc_{Po212}}{t} \frac{36}{64} \right) \right] \frac{C_p \, C_T}{(\varepsilon_0 - b \, [H_2O])^2} \right)^2 u_{\varepsilon_0}{}^2 +$$

$$\left( \left[ \frac{nc_{Po218}}{t} - \left( \frac{nc_{Po212}}{t} \frac{36}{64} \right) \right] \frac{C_p \, C_T \, [H_2O]}{(\varepsilon_0 - b \, [H_2O])^2} \right)^2 u_b{}^2 + \left( \left[ \frac{nc_{Po218}}{t} - \left( \frac{nc_{Po212}}{t} \frac{36}{64} \right) \right] \frac{C_p \, C_T \, b}{(\varepsilon_0 - b \, [H_2O])^2} \right)^2 u_{[H_2O]}{}^2 + \left( \left[ \frac{nc_{Po218}}{t} - \right. \right.$$

$$\left. \left. \left( \frac{nc_{Po212}}{t} \frac{36}{64} \right) \right] \frac{C_T \, P_{ref}}{\varepsilon \, P^2} \right)^2 u_P{}^2 + \left( \left[ \frac{nc_{Po218}}{t} - \left( \frac{nc_{Po212}}{t} \frac{36}{64} \right) \right] \frac{C_P}{\varepsilon \, T_{ref}} \right)^2 u_T{}^2 \quad (10)$$

Table 1 presents the different contributions to the total uncertainty of each radon measurement performed with the ARMON v2. In this example the average radon concentration, water vapour concentration, hourly [212]Po counting, atmospheric Pressure
and Temperature from a 6 months intercomparison within the traceRadon project at Saclay Atmospheric Station (SAC) were selected as reference values to perform an estimation. Integration time for radon concentration measurement was of 1 h.

| Quantity | Estimate | Type | Standard uncertainty | Probability distribution | $v_i$ | Sensitivity coefficient | Contribution to the standard uncertainty |
|---|---|---|---|---|---|---|---|
| $X_i$ | $x_i$ | | $u(x_i)$ | | | $c_i$ | $u_i(y)$ |
| $nc_{Po218}$ | $nc_{Po218}$ | A | $\sqrt{nc_{Po218}}$ | Normal | $\infty$ | $\dfrac{C_p \, C_T}{t \, F_{cal}}$ | $c_i u(x_i)$ |
| $nc_{Po212}$ | $nc_{Po212}$ | A | $\sqrt{nc_{Po212}}$ | Normal | $\infty$ | $-\dfrac{C_p \, C_T}{t \, F_{cal}} \dfrac{36}{64}$ | $c_i u(x_i)$ |
| $\varepsilon_0$ | 0.0057 (Bq m$^{-3}$)$^{-1}$ s$^{-1}$ | B | 0.01 (3%) [1] | Normal | $\infty$ | $-\left[ \dfrac{nc_{Po218}}{t} - \left( \dfrac{nc_{Po212}}{t} \dfrac{36}{64} \right) \right] \dfrac{C_p \, C_T}{(\varepsilon_0 - b \, [H_2O])^2}$ | $c_i u(x_i)$ |
| $b$ | 5.4 10$^{-7}$ (Bq m$^{-3}$)$^{-1}$ s$^{-1}$ ppmv$^{-1}$ | B | 7.3 10$^{-8}$ [2] | Normal | $\infty$ | $\left[ \dfrac{nc_{Po218}}{t} - \left( \dfrac{nc_{Po212}}{t} \dfrac{36}{64} \right) \right] \dfrac{C_p \, C_T \, [H_2O]}{(\varepsilon_0 - b \, [H_2O])^2}$ | $c_i u(x_i)$ |
| $[H_2O]$ | ~254 ppmv | B | 20% $[H_2O]$ + 1ppmv [3] | Normal | $\infty$ | $\left[ \dfrac{nc_{Po218}}{t} - \left( \dfrac{nc_{Po212}}{t} \dfrac{36}{64} \right) \right] \dfrac{C_p \, C_T \, b}{(\varepsilon_0 - b \, [H_2O])^2}$ | $c_i u(x_i)$ |
| $P$ | ~1000 hPa | B | 0.3 hPa [4] | Normal | $\infty$ | $-\left[ \dfrac{nc_{Po218}}{t} - \left( \dfrac{nc_{Po212}}{t} \dfrac{36}{64} \right) \right] \dfrac{C_T \, P_{ref}}{\varepsilon \, P^2}$ | $c_i u(x_i)$ |
| $T$ | ~298 K | B | 0.15 + 0.002*T [3] | Normal | $\infty$ | $\left[ \dfrac{nc_{Po218}}{t} - \left( \dfrac{nc_{Po212}}{t} \dfrac{36}{64} \right) \right] \dfrac{C_P}{\varepsilon \, T_{ref}}$ | $c_i u(x_i)$ |
| $C_{Rn}$ | Eq- (9) | Combined uncertainty (u) (Bq m$^{-3}$) | | | | | $u = \sqrt{\sum u_i^2(y)}$ |

[1]Uncertainty from the calibration at INTE Radon Chamber
[2]Residual -Standard Error from correlation linear model according to calibration at INTE radon chamber.



[3]*From manufacturers documentation*
       [4]*From ICOS  Atmosphere Station specification, v2.0 ([https://box.lsce.ipsl.fr/index.php/s/uvnKhrEinB2Adw9?path=%2FSpecifications](https://box.lsce.ipsl.fr/index.php/s/uvnKhrEinB2Adw9?path=%2FSpecifications))*

**Table 1: Contributions of the different variable and/or parameters to the total uncertainty of a typical radon concentration measurement performed with the ARMON v2 at an atmospheric station.**


As the acquisition chain of the ARMON v2 allows to separate the energy of the alpha particles emitted by the different Polonium isotopes, thus [210]Po counts can be skipped and it will be not influencing the instrument background. Interference to the [218]Po counts are only due to [212]Bi as it was explained in section 2.1. Therefore, the typical limits (threshold limit and detection limit) will depend on the presence of thoron within the sampled air.

According to the ISO 11929-4, the decision threshold of the activity ($a^*$) can be calculated using Eq. (11):

$$a^* = k_{1-\alpha}\, \tilde{u}(0) \;=\; k_{1-\alpha}\, \sqrt{w^2 \left( \frac{n_0}{t_g t_0} + \frac{n_0}{t_0^2} \right)} \ (11)$$

where $k_{1-\alpha}$= 1.645, $\tilde{u}(0)$ is the standard uncertainty of the background, $w$ is the calibration factor ($1/\varepsilon$), $n_0$ is the number of counts of the background effect, and $t_0$ and $t_g$ are the count times of the measurement and the background.

The detection limit, according to the same standard, can be calculated, with a 95 % confidence, as in in Eq. (12)

$$a^\# = \frac{2\,a^* + (k^2\,w)/t_g}{1 - k^2\,u_{rel}^2(w)} \ (12)$$

being the $u_{rel}(w)$ relative standard uncertainty of the estimated efficiency $\varepsilon$.

## 2.5 Evaluation of the linearity of the ARMON v2 detection efficiency for low radon concentrations

The linearity of the detection efficiency of the ARMON v2 was checked thanks to the availability of a new methodology, developed within the WP1 of the traceRadon project too, to create low radon reference atmosphere of few Bq m$^{-3}$ using low

radon emanation sources developed by radioactivity group of the PTB (Röttger et al., 2023). The ARMON v2 was actually exposed within the climatic chamber of the PTB (see Appendix B, Fig. B1) under radon levels of few tens of Bq m$^{-3}$and during several months.

The PTB chamber has a nominal volume of $V = (21.035 \pm 0.030)$ m$^3$, which makes a calibration of larger devices inside the

chamber possible. This chamber is equipped with a walkable air lock system and can be operated in a temperature range between -20 °C and +40 °C, as well as between 5 % to 95 % relative humidity. The pressure inside the chamber is recorded. The walls of the chamber consist of 100 mm polyurethane foam, clad inside and outside with stainless steel 0.6 mm in thickness. Due to this construction, the heat transmission coefficient is smaller than $k = 0.2$ W m$^{-2}$ K$^{-1}$, which provides very stable calibration conditions. The inner wall is polished and connected to the ground, thus providing a homogeneous radon

progeny field (Honig et al., 1998). Within the chamber the traceable $^{222}$Rn activity concentration is established either via a $^{222}$Rn gas standard (Dersch and Schötzig, 1998) or via primary $^{226}$Ra emanation sources (Mertes et al., 2022). Due to the low activity concentrations values intended during this calibration (5 Bq m$^{-3}$ to 20 Bq m$^{-3}$) the emanation source technique was used (Röttger et al., 2023). A $^{222}$Rn free background was achieved, applying aged, synthetic, compressed air to the chamber, flushing all remainders of $^{222}$Rn from it.

Extensive experiment over a period of 4 months with varying activity concentrations between $(7.8 \pm 0.4)$ Bq m$^{-3}$ and $(45.4 \pm 0.8)$ Bq m$^{-3}$ have been carried out. Even though dry air had been applied through the background determination, additional silica gel and a thoron delay volume were installed at the inlet of the ARMON v2, to prevent thoron progeny events and humidity during the experiment. All installations and detectors were completely installed inside the climate chamber, which was operated in a closed mode, to prevent any exchange with the surrounding low activity concentration lab air. All

results are in consistence with this assumption.





### 3 Results and discussion

#### 3.1 Theoretic efficiency

The EF and its force lines inside the sphere, when $V = 10$ kV was applied, was modelled with the COMSOL software, and are shown in Fig. 2a. The simulation of the tracks of 10 000 randomly spaced particles using this EF (Figures. 2b, 2c and 2d)

shows that the 98 % of the $^{218}$Po$^+$ particles generated inside the spherical detection volume are collected inside the active area of the detector if we assume no interactions with other particles, decay or neutralisation. Applying Eq. (4), and assuming that $\%_{218_{Po^+}}$=0.88, $\%_{Active}$= 0.98 and $\%_{Detected}$=0.5, the maximum efficiency of our geometry, $\varepsilon_g$, in terms of counts detected per disintegrations inside the detection volume will be of 43 %. If we express the efficiency in terms of count rate (s$^{-1}$) per Bq m$^{-3}$, assuming a detection volume of 0.02 m$^{-3}$, the $\varepsilon_g$ efficiency of our system is 0.0086 (Bq m$^{-3}$) $^{-1}$ s$^{-1}$.

From the simulation of the trajectories of the 10 000 polonium ions, the estimated travelling time of the particles to reach the detector surface will vary between 0 s and 1.8 10$^{-2}$ s, depending on its distance from the detector, with a mean value of 8.9 10$^{-3}$ s. During these travelling times, the probability of $^{218}$Po decay events will be completely negligible, while the effect of the recombination with OH$^-$ particles will cause a loss of particles from 0 % to 25 % in an interval between 0 ppmv and 2000 ppmv. Consequently, the collection efficiency $\varepsilon_c$ will vary between 100 % at 0 ppmv and 75 % at 2000 ppmv, being

87.6 % at the nominal humidity of 400 ppmv.

Multiplying both geometrical and collection efficiencies, the maximum theoretical efficiency of our system, when no water is present, will be $\varepsilon_0 = 0.0086$ (Bq m$^{-3}$)$^{-1}$ s$^{-1}$, while when working at 400 ppmv and 2000 ppmv of H$_2$O the theoretical $\varepsilon$ will be 0.0075 (Bq m$^{-3}$)$^{-1}$ s$^{-1}$ and 0.0065 (Bq m$^{-3}$)$^{-1}$ s$^{-1}$ respectively. Figure 3 shows the relationship between the estimated theoretical detection efficiency of the ARMON v2 in relation to the water content of the sampled air (blue marine line).


It should be take into account that during the simulations some hypothesis were done which may be not completely consistent with the reality: i) no others recombination processes of the $^{218}$Po particles were considered; ii) a regular spherical potential surface was considered to generate an EF with spherical symmetry although the real EF is expected to have some irregularities due to the inhomogenous distribution of the potential over the sphere wall due, among others, to the presence of inlet and outlet

tubbing connections; iii) no air diffusion effects were considered, iv) it has been observed in the results of the COMSOL simulations that a small vertical shift in the detector position could change the percent of particles collected on the active area of the detector surface All these previous observations lead to the conclusion that the theoretical efficiency obtained for the ARMON v2 has only to be considered as a the ideal highest value and not treated as nominal efficiency of the instrument.





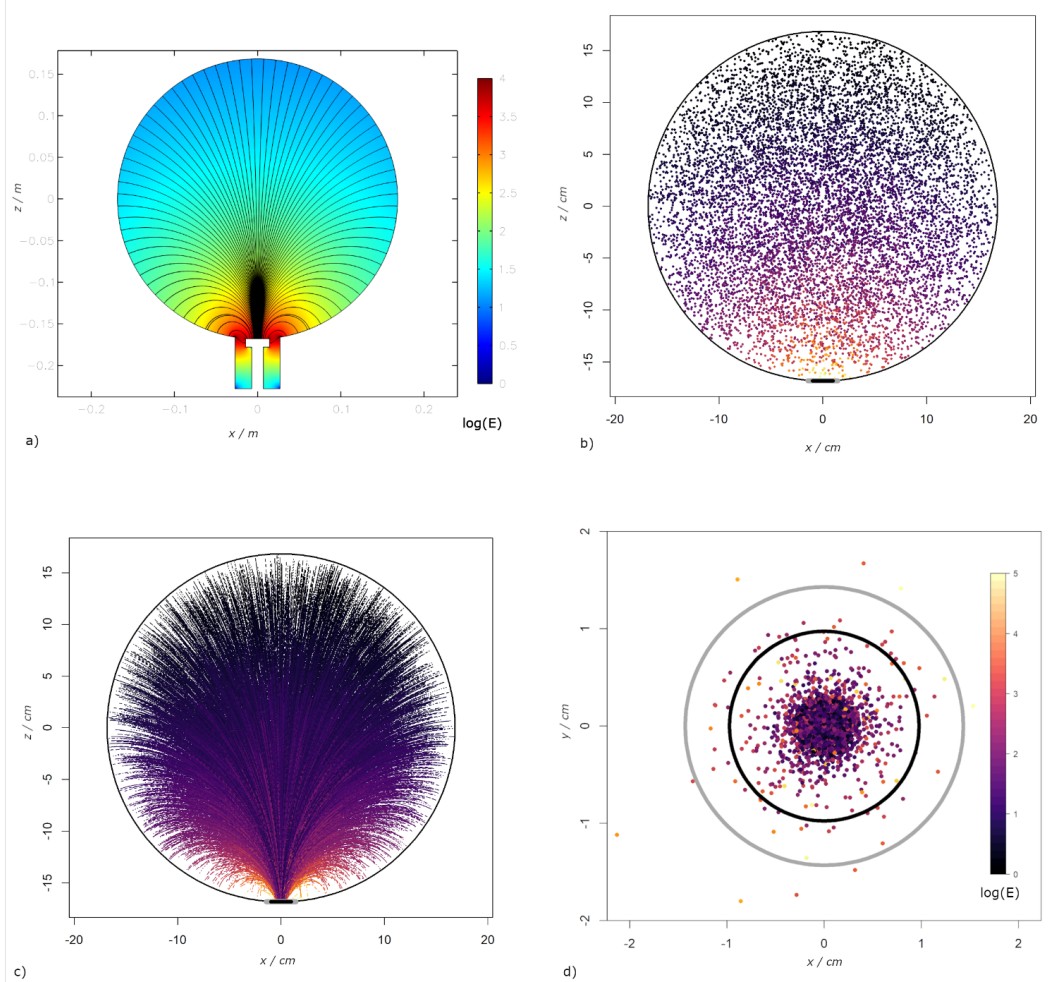

**Figure 2. a): Simulation of the electrostatic field generated within the ARMON v2 detection volume with the application of 10 kV voltage, black lines represent the EF direction; b): Initial position inside the detection volume of the simulated $^{218}$Po ions ($10^5$ fictitious particles); c): Trajectories of the simulated particles inside the sphere when the 10 kV voltage is applied between the sphere walls and the PIPS detector; d): Distribution of the simulated deposition of the charged particles at the detector surface. Inner black circle denotes the active area. Colour scale for Fig. 2a, 2b and 2c is the common logarithm of the EF, in log(V/cm), and it is shown** 350 **in Fig. 2d.**

### 3.2 INTE Calibration results

Figure 3 shows the results of the water correction experiments carried out at the INTE-UPC and PTB chambers. A linear relationship between the detection efficiency of the instrument and the water vapour concentration is observed within a range of 150 ppmv - 2000 ppmv. This relationship was found to be independent on the radon concentration in a range of 600 Bq m$^{-3}$ 355 $^{-3}$ - 5900 Bq m$^{-3}$. When the water vapour concentration of the sampled air is above 2000 ppmv the relation loses this linearity and for this reason it is worth not to measure over this vapour concentration. In the range 150 ppmv - 2000 ppmv, the detection efficiency of the ARMON v2 may be corrected using Eq. (2) with *b* being equal to 5.4 10$^{-7}$ (Bq m$^{-3}$)$^{-1}$ s$^{-1}$ ppmv$^{-1}$ with an uncertainty (RSE) of 7.3 10$^{-8}$ (Bq m$^{-3}$)$^{-1}$ s$^{-1}$ ppmv$^{-1}$.




Due to the increment observed in the detection efficiency for values of water vapour concentration lower than 150 ppmv, an exponential correction fit was also applied to the data following Eq. 13.

$\varepsilon = \varepsilon_0' e^{(b' \, [H_2O]^{1/2})}$ (13)

The exponential curve (green dashed line) is also represented in Fig. 3 and may be more appropriate for very low water concentrations which are usually uncommon for sampled air at atmospheric stations. For this reason, the use of the linear fit is here proposed.

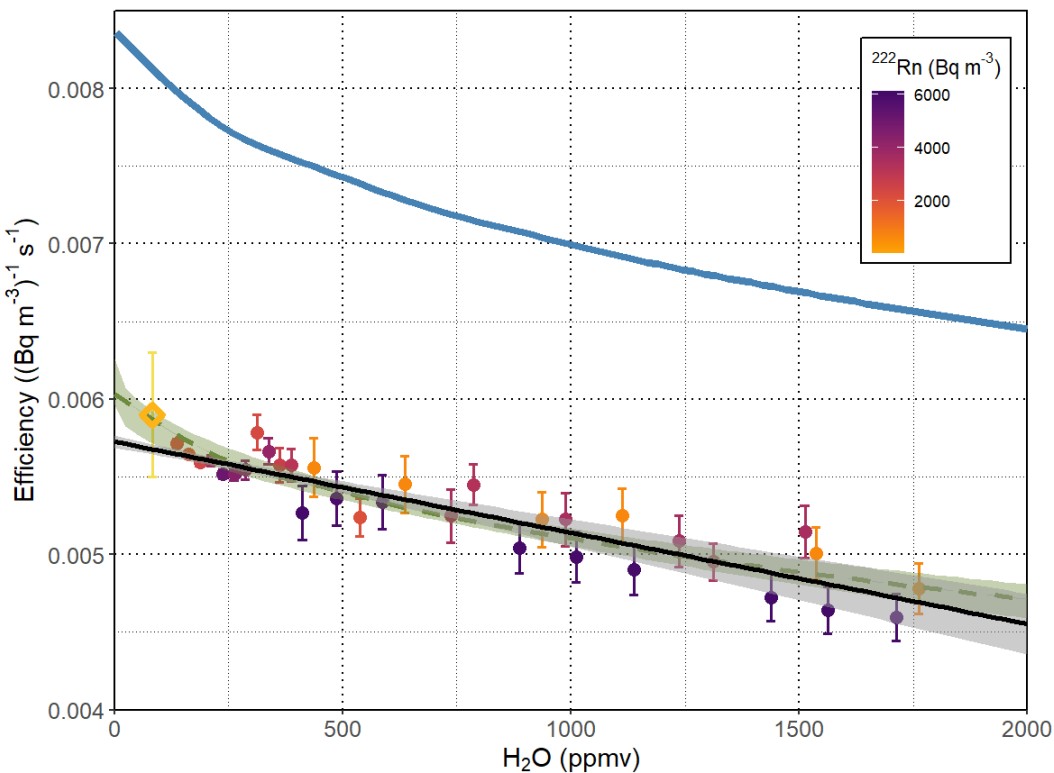

**Figure 3: Dependence of the efficiency of the ARMON v2 monitor on to the water vapour concentration (in ppmv H₂O) at the detection volume. The coloured points are efficiency averages and its uncertainty in intervals of 10 ppmv of H₂O, of the efficiency of the hourly measurements for all the calibrations at INTE-UPC. The black line is the linear fit of the observational points with the 95 % confidence interval represented by the grey shaded zone. The green dotted curve is the exponential fit of the observational points with the 95 % confidence interval represented by the green shaded zone. The rhomboid represents the efficiency of the**
**ARMON at PTB with its uncertainty. The blue curve represents the theoretical efficiency simulation assuming a mobility of 3 cm² (V s)⁻¹.**

Once determined the water correction coefficient $b$, the efficiency of the monitor $\varepsilon_0$ was calculated within the radon concentration range of 500 Bq m⁻³ - 6000 Bq m⁻³. From the results obtained (Fig. 4), a high linearity ($r^2 = 0.999$) in the
regression between ²¹⁸Po counts against ²²²Rn concentration measured with the ATMOS monitor was observed. Within the calibration range (300 Bq m⁻³ - 6200 Bq m⁻³), and taking in consideration the ATMOS uncertainty, the $\varepsilon_0$ of the ARMON v2 calculated with the ATMOS monitor at the INTE chamber was of (0.0057 ± 0.0002) (Bq m⁻³) s⁻¹.

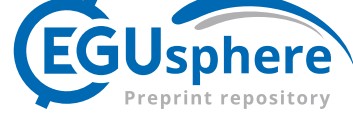



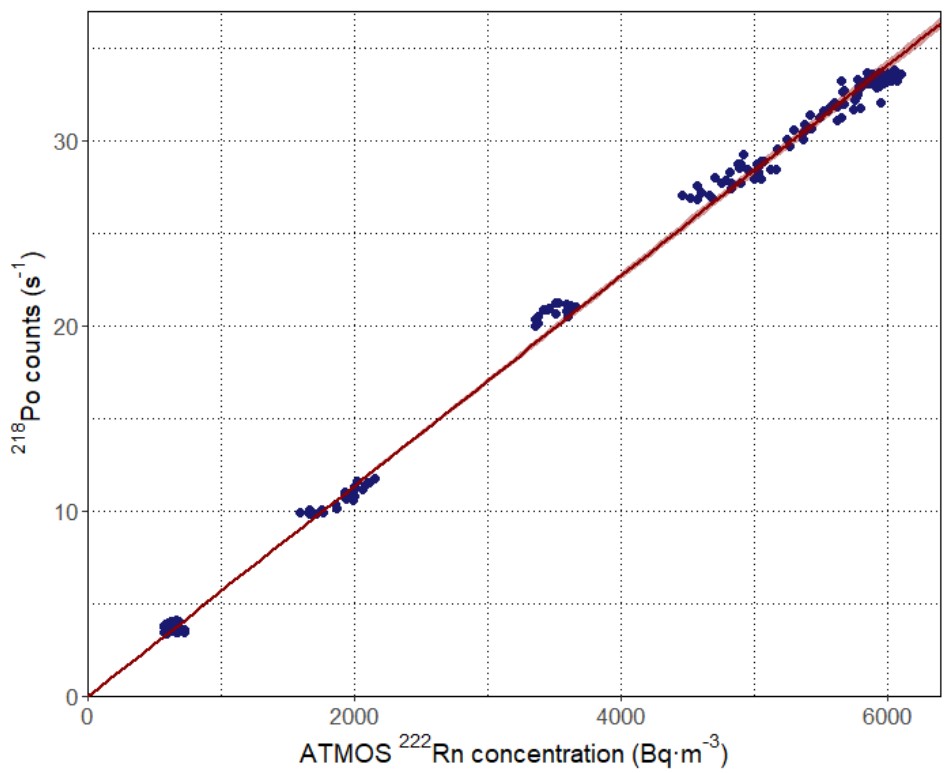

**Figure 4: Calibration of the efficiency of the ARMON v2 monitor ($^{218}$Po counts against $^{222}$Rn concentration) within the range 0**
**Bq m$^{-3}$ – 6000 Bq m$^{-3}$. $^{222}$Rn concentration measured with an ATMOS monitor at the INTE-UPC radon chamber (hourly means).**
**$^{218}$Po counts (s$^{-1}$) from hourly spectra. Red line is the regression line ($r^2 = 0.999$).**

It has to be underlined that the experimental calculated efficiency of the ARMON v2 at the range between 300 ppmv -

2000 ppmv of $[H_2O]$ is a 24 % lower than the theoretical one (assuming a mobility of 3 cm$^2$ (V s)$^{-1}$). Although in the same

order of magnitude, this difference could be explained, as described in section 3.1, by a multitude of variables which could

cause the $^{218}$Po ions not to be collected at the detector surface.

### 3.3 Uncertainty, background and typical limits

The total uncertainty of the radon measurements performed with the ARMON v2 is calculated with Eq. (9). As example here

it has been estimated for a typical atmospheric hourly radon measurement performed at the SAC atmospheric site ($C_{Rn} =$

4 Bq m$^{-3}$, $T$ = 298 K, $P$ = 1000 hPa, $[H_2O]$ = 250 ppmv and $nc_{Po212}$ = 1). The uncertainty values for all parameters and its

sensitivity coefficients are shown in Table 2. The combined uncertainty obtained was 0.46 Bq m$^{-3}$, a 11 % of the absolute value

of the measurement. The most influencing contribution in the calculation of the total uncertainty of the measurement is the

uncertainty of the total net $^{218}$Po counts, followed by the uncertainty of the detection efficiency and the uncertainty of the water

vapour correction factor.

Calculating the variability for a range of humidity (0 ppmv - 2000 ppmv), the total uncertainty of the measure has been plotted

as a function of radon concentration (Fig. 5a). In the range of 0 ppmv – 400 ppmv, the total uncertainty is below the 10 % for

radon concentrations greater than 5 Bq m$^{-3}$. For humidity greater than 1000 ppmv, the uncertainty increases due to the decrease

of the detection efficiency.





| Quantity | Estimate | Type | Standard uncertainty | Probability distribution | $v_i$ | Sensitivity coefficient | Contribution to the standard uncertainty |
|---|---|---|---|---|---|---|---|
| $X_i$ | $x_i$ | | $u(x_i)$ | | | $c_i$ | $u_i(y)$ |
| $nc_{Po218}$ | 81 | A | 9 | Normal | $\infty$ | 0.0496 | 0.4466 |
| $nc_{Po212}$ | 1 | A | 1 | Normal | $\infty$ | $-0.0279$ | -0.0279 |
| $\varepsilon_0$ | 0.0057 (Bq m$^{-3}$)$^{-1}$ s$^{-1}$ | B | 0.01 | Normal | $\infty$ | $-11.671$ | -0.1225 |
| $b$ | 5.4 10$^{-7}$ (Bq m$^{-3}$)$^{-1}$ s$^{-1}$ ppmv$^{-1}$ | B | 7.3 10$^{-8}$ | Normal | $\infty$ | 2917.7 | 0.0117 |
| $[H_2O]$ | ~250 ppmv | B | 51 .8 | Normal | $\infty$ | 3.73 10$^{-4}$ | 0.0190 |
| $P$ | ~1000 hPa | B | 0.3 | Normal | $\infty$ | $-4.00$ 10$^{-3}$ | -0.0012 |
| $T$ | ~298 K | B | 0.746 | Normal | $\infty$ | 1.1339 10$^{-2}$ | 0.0100 |
| $C_{Rn}$ | 4.0 Bq m$^{-3}$ | | Combined uncertainty (u) (Bq m$^{-3}$) | | | | 0.46 |

**Table 2. Calculated contributions of the different variable and/or parameters to the total uncertainty of a typical radon concentration measurement performed with the ARMON v2 at an atmospheric station.**

In addition, given a typical water content in sampled air of 250 ppmv H$_2$O, the total uncertainty of the measurement has been also calculated taking into account different possible levels of thoron gas in the sample (Fig. 5b). It can be observed that when

the radon concentration increases to tens of Bq m$^{-3}$, the thoron concentration present in the sampled air has almost no effect on the uncertainty of the measurement. However, at low radon concentrations below 5 Bq m$^{-3}$, the thoron concentration can be an important source of uncertainty. This problem can be easily skipped using a thoron decay volume before the ARMON v2 detection volume.

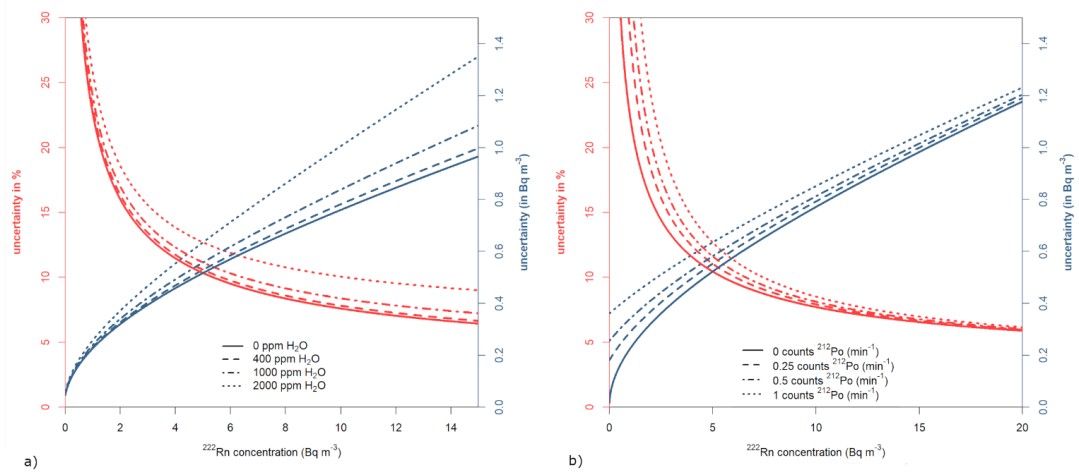

**Figure 5: a): absolute (blue) and relative (red) uncertainty as a function of $^{222}$Rn activity concentration at different water vapour concentrations. b): absolute (blue) and relative (red) uncertainty as a function of $^{222}$Rn activity concentration at different $^{212}$Po (thoron decay) concentrations.**

As an additional information, it may be of interest to explain that during the INTE-UPC experiments it was discovered that

the silica gel material may contain thorium material which is a thoron source. Actually, hourly spectra showed up to 1 count per minute (min$^{-1}$) of $^{212}$Po, which means 0.56 counts (min$^{-1}$) of $^{212}$Bi α-decays to $^{208}$Tl and this implies an increase greater than 50 % of the uncertainty for radon concentrations below 5 Bq m$^{-3}$. For this reason, and although the content of thorium



material within commercial silica gel has not yet been quantitatively estimated, authors highly recommend to do not use this dryer for radon measurements or using a delay volume of at least 10 L between the Silica Gel dryer and the selected radon

instrument. Generally, authors suggest the use of other drying systems as Nafion tubes or cold traps.

In regard to the detection limit and the decision threshold of the ARMON v2, these previous values are only dependent on the presence of thoron concentrations within the detection volume. When no thoron counts are present (e.g. when using a buffer volume before the ARMON v2), the decision threshold is 0 Bq m$^{-3}$ and the detection limit is $a^{\#}$= 0.132 Bq m$^{-3}$. At a typical

thoron concentration at atmospheric sites (100 m tall towers) of 0.017 min$^{-1}$, the detection limit and the decision threshold are 0.3 Bq m$^{-3}$ and 0.08 Bq m$^{-3}$ respectively. The change of the characteristic limits as a function of the $^{212}$Po detected count rate in min$^{-1}$ is shown in Fig. 6.

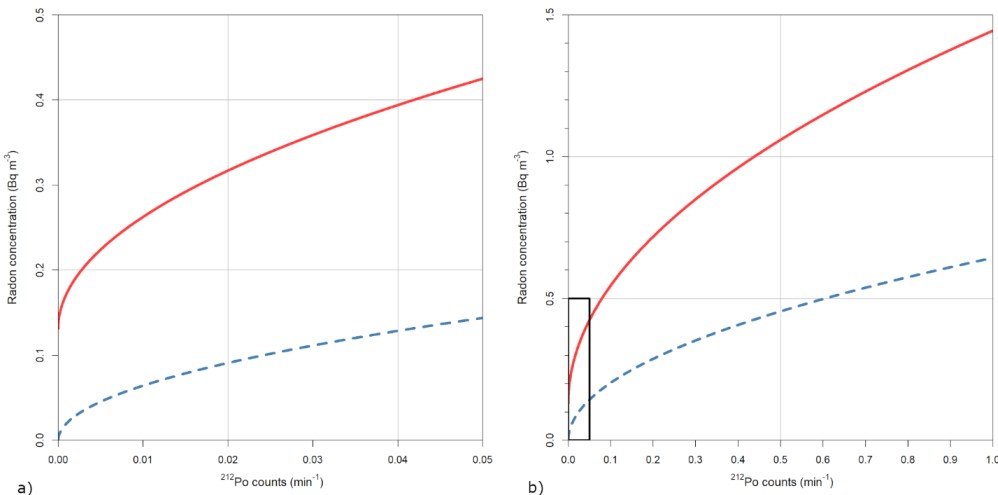

**Figure 6: Radon activity concentration detection limit (red straight) and decision threshold (dashed blue) of the ARMON v2 monitor from a) 0 counts to 0.05 counts (min$^{-1}$) and b) 0 counts to 1 count (min$^{-1}$) of $^{212}$Po detected.**

### 3.4 PTB results

Figure 7 shows a summary of the results of the values of the detection efficiency of the ARMON v2 obtained by INTE-UPC (orange dots) and PTB (blue dots) experiments. Both experiments, carried out under different conditions of radon

concentrations, show a linearity in the counts detected by the instrument and the radon concentrations to be measured (Figure 7a). Totally, five calibration points with three different emanation sources were realised at the PTB (see Appendix B Fig. B2).

During the ARMON v2 exposures at the PTB climate chamber, no variation of the humidity was investigated and the sampled air had an average water content of 83 ± 21 ppmv, during the whole measurement campaign (Fig. 3, gold rhombus) and the

estimated detection efficiency was corrected applying Eq 2.

Since five calibration points with three different emanation sources were realised (see Appendix B Fig. B2) a) and the characterisation of the sources had been done with the same instruments, the correlation of the sources and their influence on the resulting uncertainty has to be further investigated in detail. A full correlation of the sources and their uncertainties was considered at this point, which probably overestimates the total uncertainty of the calibration and increases the uncertainty

about a factor of two, with respect to just ignoring the correlation.





Taking all this into account, the sensitivity of the ARMON v2 that was determined during the calibration described in section 2.5 is $(0.00595 \pm 0.0008)$ $(Bq\ m^{-3})^{-1}\ s^{-1}$. This result is in good agreement with the one obtained from INTE-UPC exposures as previously reported in 3.2. The offset determined during this calibration is with $(0.002 \pm 0.007)\ s^{-1}$ in good agreement with the theoretical 0.

The detection efficiency of the ARMON v2, within its uncertainty, do not change when the radon concentrations vary between few $Bq\ m^{-3}$ and thousands of $Bq\ m^{-3}$ (Figure 7b). This is an important output which confirms the robustness of this instrument and its response. This last result also allows to accept and to use the detection efficiency value obtained at high radon concentrations and for this reason with a much smaller uncertainty. Additionally, the stability of the linearity in time and in a wide range of radon concentrations of the detection efficiency of the ARMON v2, proofs its suitability to be used as a transfer standard for in situ calibration and/or comparison of others radon and radon progeny monitors.

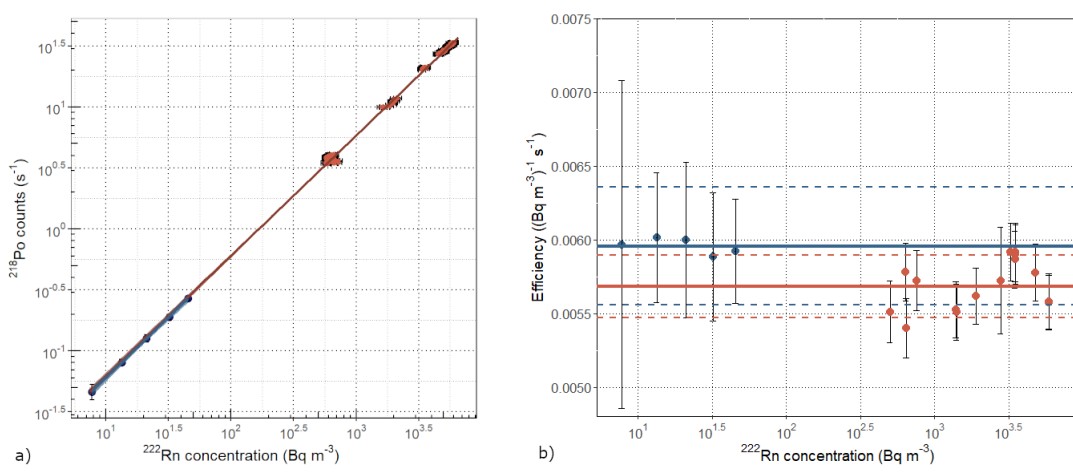

a)

b)

**Figure 7: a) Counts per second versus radon concentration (dots) and regression lines for the detection efficiency obtained during INTE-UPC experiments (orange) and PTB experiments (blue), with the 99 % confidence level shadowed. b) Dots: detection efficiency of the ARMON v2 and its uncertainty versus radon concentration for the different exposures at PTB (blue) and INTE-UPC (orange). Solid lines are the mean of the efficiency values obtained at PTB (blue) and INTE (orange), with its uncertainty at $k=1$ (dashed lines). X axis for both figures and Y axis for figure 7a are in logarithmic scale.**

It needs to be underlined that, if the exponential correction was used for the water vapour conditions (Fig. 3), the detection efficiency obtained by INTE-UPC and PTB experiments will be even more similar for both calibrations, as $\varepsilon'_0$ will be estimated to 0.0061 and 0.0062 for INTE-UPC and PTB calibrations, respectively.

## 4 Summary, conclusions and further steps

In this paper, a new version of the Atmospheric Radon MONitor (ARMON) is described. This new version is more robust and transportable than the previous prototype, can be easily installed at atmospheric stations and can be remotely controlled thank to a GUI window.

For the first time ever, the response of the ARMON v2 has been fully characterized by both theoretical and experimental approaches to obtain its detection efficiency for different radon concentrations, spanning between few $Bq\ m^{-3}$ and thousands of $Bq\ m^{-3}$. A total uncertainty budget of the ARMON v2 monitor has been also carried out for the first time. Independent



experiments were carried out both at the INTE-UPC radon chamber and at the PTB climate chamber in the framework of the European project traceRadon.


The monitor detection efficiency was found to be $(0.0057 \pm 0.0002)$ (Bq m$^{-3}$)$^{-1}$ s$^{-1}$ according to the INTE-UPC exposures results, and of $(0.00595 \pm 0.0008)$ (Bq m$^{-3}$)$^{-1}$ s$^{-1}$ according to the PTB experiments. The combined uncertainty of the ARMON v2 is lower than 10 % for radon activity values higher than 5 Bq m$^{-3}$ and the detection limit 0.132 Bq m$^{-3}$ when no thoron concentration is present in the sampled air. The theoretical detection efficiency was of $(0.0075$ (Bq m$^{-3}$)$^{-1}$ s$^{-1}$), which is

a 27 % higher than the real one, assuming that there are factors that where not taken into account as possible irregularities of the Electrostatic Field or recombination od $^{218}$Po$^{+}$ ions with other particles.

The linearity of the ARMON v2 response observed thanks to the INTE-UPC and PTB experiments allows the instrument to be calibrated at high concentration values and thus to reduce the calibration uncertainty.


In addition to the present full characterization of the ARMON v2, another completely different calibration method based on short pulse of $^{222}$Rn was applied at PTB in the framework of the same traceRadon project. Due to the special features of the ARMON v2 detector, this will allow for very short calibration or recalibration, also outside a calibration chamber and under field conditions.  Results are still under investigation and will be the object of a future paper. Finally, the ARMON v2 was also

compared under field conditions with the new ANSTO 200 L (Chambers et al., 2022) and its results will be published in third scientific paper.

From the results of the present study, it can be confirmed that the ARMON v2 can be considered a good transfer standard for in situ calibration of radon and radon progeny monitors installed at atmospheric sites according to the requirements of the

atmospheric radon community.

**Appendix A. ARMON v2 supplementary figures**

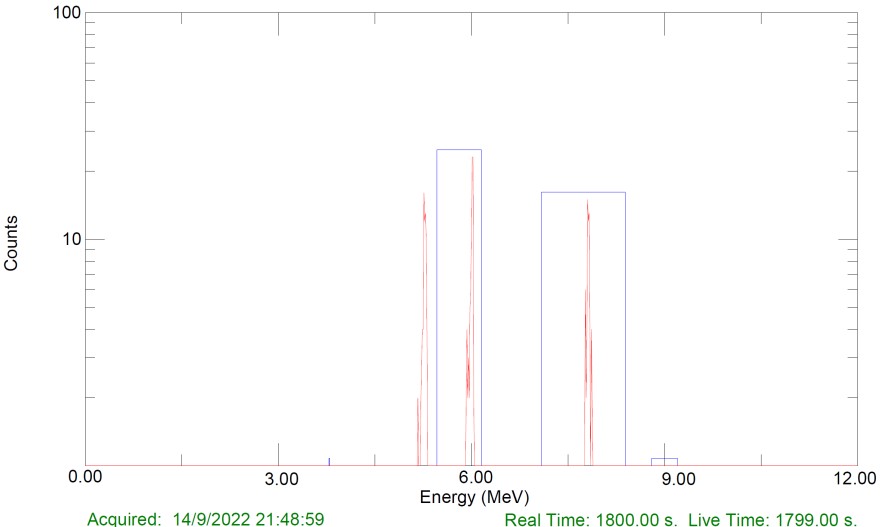

**Figure A1 . Typical spectrum from the ARMON monitor with the 210Po (5.30 MeV), 218Po (6.0 MeV) and 214Po (7.69 MeV) peaks**
**observed. No 212Po and 214Po counts are observed.**



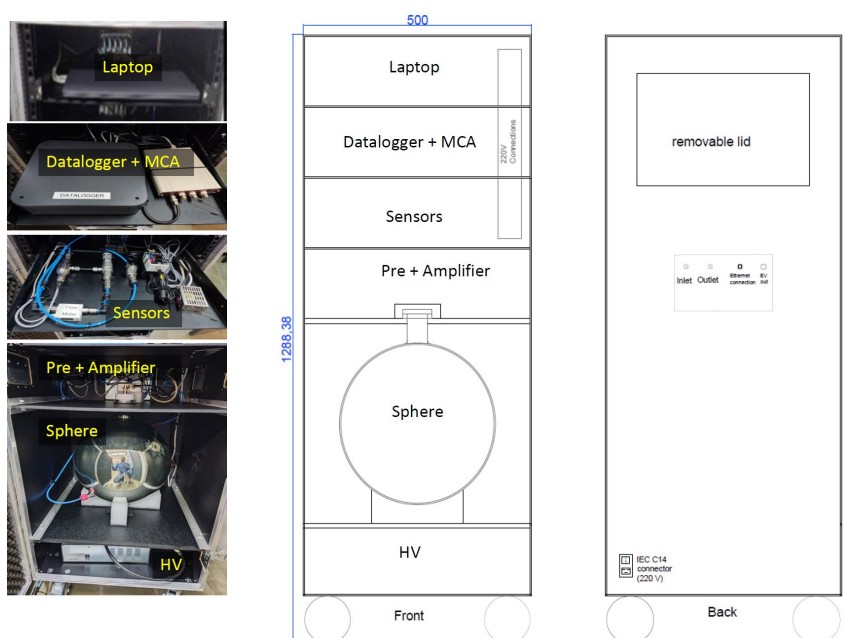

Figure A2. ARMON monitor. Left: Trays and parts. Middle and *r*ight: inside and back drawing.


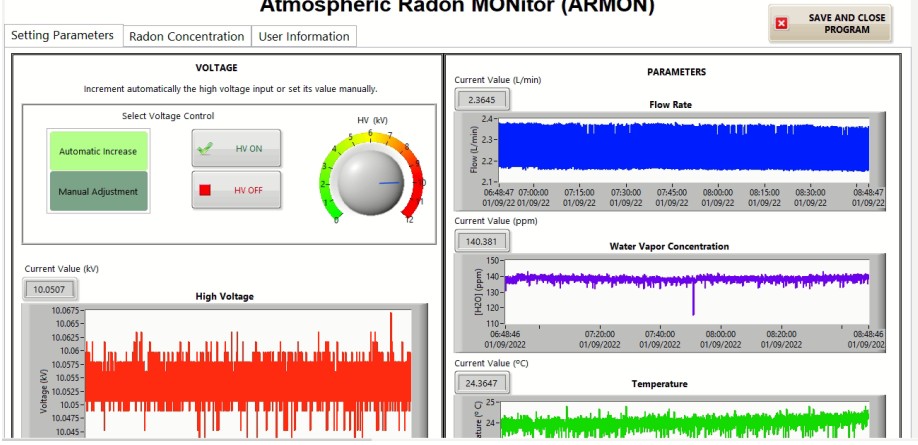

a)





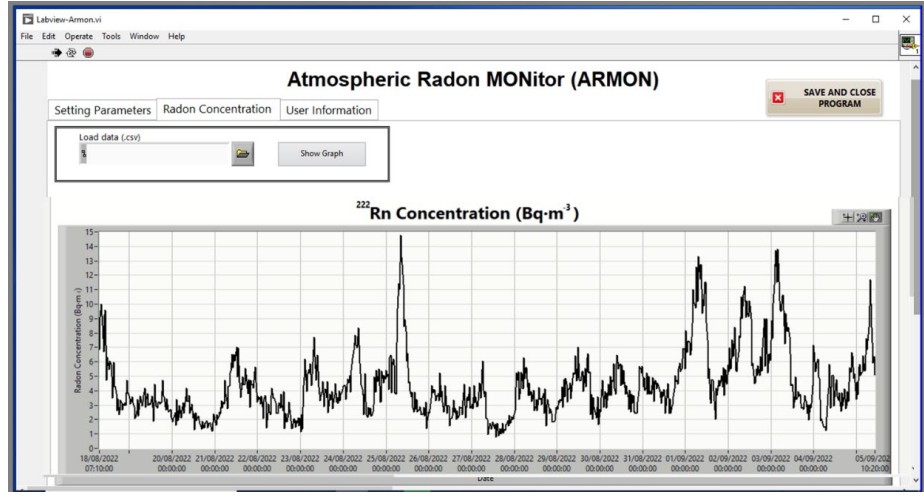

b)

**Figure A3. User interface of the new ARMON monitor. a) Sensor and voltage control b) Radon concentration visualization tab.**

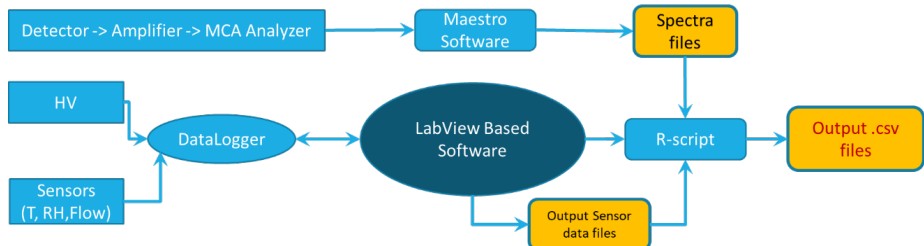

**Figure A4. Data flow chart of the ARMON v2.**






**Appendix B. Calibration at PTB climatic chamber supplementary figures**

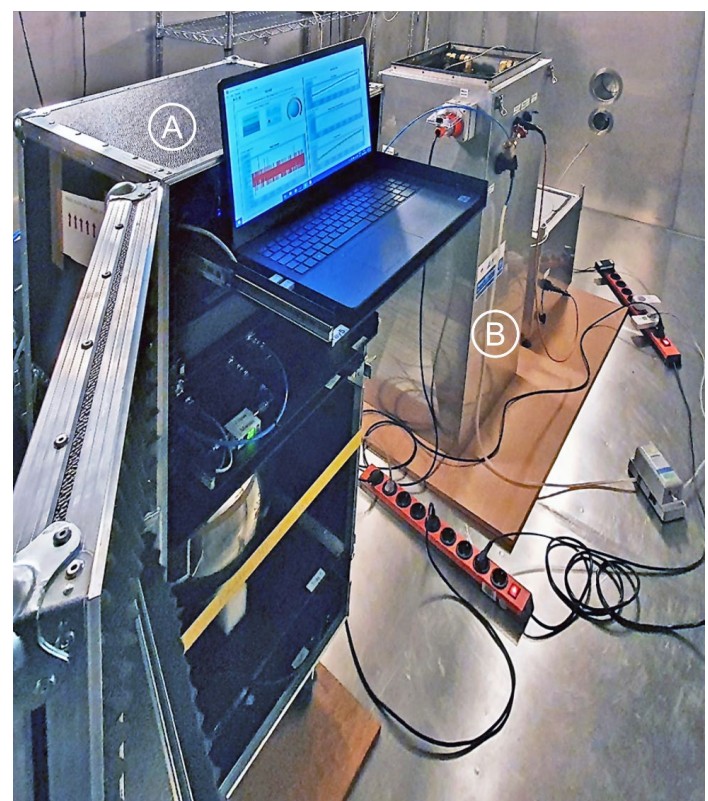

**Figure B1: Picture of the calibration setup of the ARMON v2 in the calibration chamber at PTB. In the foreground you see the opened case of the ARMON v2 (A) and in the background a monitoring system developed by ANSTO (B) (Chambers et al., 2022).**


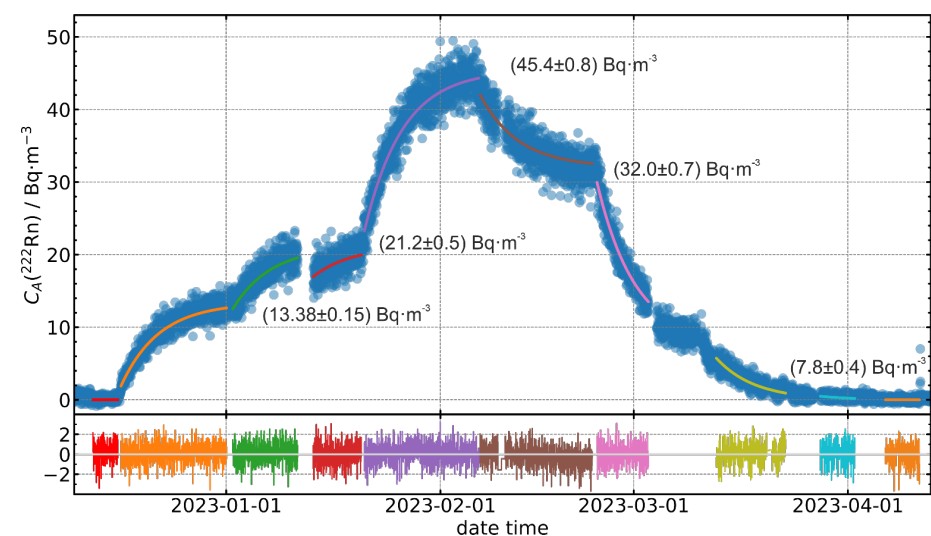

**Figure B2: Radon activity concentration determined with the ARMON v2 using three different emanation sources in five combinations. The values given in the figure illustrate the equilibrium activity concentration reached after infinite time with this source combination. The blue dots show the measured results of the ARMON v2 acquired during 30 min per point. The coloured**



**lines show the modelled activity concentration determined from the emanation sources combination. The respective coloured lines in the lower graph show the relative residual between model and measurement, which proofs the excellent agreement.**

## Code availability

The data and codes for this paper are available at the CORA Repositori de Dades de Recerca with doi
https://doi.org/10.34810/data893,

## Data availability

The data and codes for this paper are available at the CORA Repositori de Dades de Recerca with doi
https://doi.org/10.34810/data893,

## Author contributions

RC in the framework of his PhD project built the new version of the ARMON, ran the calibration experiments at the INTE-UPC and analysed the data with heir corresponding uncertainties. RG and CG designed the COMSOL setting simulations for the theoretical efficiency analysis. AV and CG supervised the work as PI and co-PIs of the MAREA project and senior researchers of the traceRadon project. SR carried out the ARMON v2 exposures at the PTB and the relative data analysis. RC led the manuscript writing and all authors contribute to it. All authors have read and agreed to the published version of the
paper.

## Competing interests

The authors declare that they have no conflict of interests.

## Acknowledgments

The authors want to acknowledge Annette Röttger as coordinator of the traceRadon project for her work and continuous
support. The authors also want to acknowledge Liza Shiro for her work in the developing of the ARMON-Labview program, Juan Antonio Romero for his support during the ARMON v2 building, Anja Honig and Tanita Ballé for their support during the ARMON exposures at the PTB and Florian Mertes for his interesting ideas during the development of the low radon emanation sources of the PTB.

## Financial support

This project was financed by the MAR2EA (IU68-017047) project and the 19ENV01 traceRadon project, which has received funding from the EMPIR programme co-financed by the Participating States and from the European Union's Horizon 2020 research and innovation programme.

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
