# Peer review of "Full characterization and calibration of a transfer standard monitor for atmospheric radon measurements"

_EGUsphere, 2023_

## Referee Comment (RC2)

[referee-annotated manuscript omitted]

---

## Community Comment (CC1)

Some comments on "Full characterization and calibration of a transfer standard monitor for atmospheric radon and thoron measurements" by Curcoll et al., currently being considered for publication in Atmospheric Measurement Techniques.

**Abstract**

- For better transparency (considering readers who may not be familiar with ISO 11929-4), it would be good in the abstract to provide the reader with context for the claimed ARMON v2 detection limit of 0.132 Bq m$^{-3}$ that would be directly comparable to other radon measurement systems. For example, some studies have shown the hourly measurement uncertainty of commercial AlphaGuard units at their nominal detection limit (of around 3 Bq m$^{-3}$) is 50 – 60% (in other cases the quoted uncertainty has been higher), and the radon concentration at which the 200 L ANSTO dual-flow-loop monitor has an hourly measurement error of 30% is around 0.14 – 0.16 Bq m$^{-3}$ (Chambers et al. 2022; doi:10.5194/adgeo-57-63-2022). Based on the results of Figure 5a of this manuscript, the hourly ARMON v2 measurement uncertainty for $^{222}$Rn in a dry, $^{220}$Rn-free environment (i.e., best case scenario) at an ambient $^{222}$Rn activity concentration of around 0.6 Bq m$^{-3}$ is ≥ 30%. Guided by the shape of the curve in Figure 5a, the hourly measurement uncertainty at the claimed detection limit of 0.132 Bq m$^{-3}$ would likely exceed 100%. Stating the hourly measurement uncertainty along with the claimed detection limit in the abstract would be a better guide for the reader.

- Furthermore, since the ARMON v2 is introduced here as being able to separately quantify radon ($^{222}$Rn) and thoron ($^{220}$Rn), it would be good to state in the abstract the expected detection limit and hourly measurement uncertainty both with, and without, the presence of $^{220}$Rn in the sampled airstream (assuming a representative $^{220}$Rn activity for the surface layer – such as the value quoted on Line 430).

- Lastly, the suitable measurement range of the ARMON v2 is quoted to be 1 – 100 Bq m$^{-3}$, but the measurement uncertainty is given only for a concentration of >5 Bq m$^{-3}$. Would it not make sense to quote the measurement uncertainty at 1 Bq m$^{-3}$? Or at least report this value also?

**Line 100:** For completeness, the authors should also consider including the following paper in this summary: Wada, A., Murayama, S., Kondo, H.,Matsueda, H., Sawa, Y., and Tsuboi, K. (2010). Development of a compact and sensitive electrostatic Radon-222 measuring system for use in atmospheric observation. J. Meteorol. Soc. Jpn. 88, 123–134. doi: 10.2151/jmsj.2010-202.

**Section 2.4:** Regarding the uncertainty and application of the STP correction for ARMON v2 measurements: according to Figure 1, the temperature measurements of the ARMON v2 are not made in the measurement sphere, but a long way downstream of the sphere and some other instruments. The location of the pressure measurements is not indicated in the figure, it would be good to see where they are made. Given the separation between the sensors and measurement volume, and the fact that the temperature sensor is in a separate, ventilated compartment of the ARMON's transport case, can the authors give any indication of the expected additional uncertainty in the derived STP correction parameter? At the moment, it seems that only the instrument manufacturer uncertainty values for temperature and pressure are being considered.

**General:** Consider revising the text for grammatical accuracy.

**Scott Chambers, ANSTO, Lucas Heights, NSW.**

---

## Community Comment (CC2)

Dear Authors,

Thank you for the response to my initial comments. I appreciate that the manuscript currently under review is not an intercomparison manuscript and I am not asking for it to become one. As the authors point out, a separate, independent characterisation and intercomparison of the new ARMON V2 with other radon monitors has recently been conducted, for which the results are still being finalised for publication. In itself, this brings into question the timing and utility of the present manuscript – given that its content may either detract substantially from the novelty of the planned independent intercomparison study or be difficult to retrospectively correct if later found to be incomplete.

The main point I was trying to make is that the transparency of reported findings is important to the research community. Every instrument has its strengths and weaknesses, and this information needs to be reported in an easily interpretable, objective manner for potential users to make an informed decision as to whether a particular instrument is fit for the purpose they intend to use it for.

Lines 31-32 claim the suitability of the ARMON V2 as a calibration transfer standard device, and Lines 50-54 imply that the main target instruments for this service would be part of the European ICOS network. Radon measurements as part of the ICOS network are typically made at heights of $100 - 200$ m above ground level. Furthermore, some of these sites are at remote, coastal or mountaintop locations, where radon concentrations are usually well below reported values typical of the terrestrial boundary layer globally (the value quoted by the authors was 5 Bq/m$^3$).

As an example, Cabauw is a key European ICOS site, located in a flat inland region. Based on 10 years of Cabauw radon observations at 200m agl, the 10$^{th}$, 50$^{th}$ and 90$^{th}$ percentile radon concentrations for this site are 0.35, 1.1 and 3.6 Bq/m$^3$, respectively. Another relevant ICOS station is Saclay, in France. Based on measurements at this site in 2022 at 100m agl the 10$^{th}$, 50$^{th}$ and 90$^{th}$ percentile radon concentrations were 1, 2.5 and 6.1 Bq/m$^3$, respectively. If the uncertainty of the ARMON V2 is only less than 10% for radon concentrations above 5 Bq/m$^3$, would the authors be able to comment on the implications for transferring an SI traceable calibration to an operating radon monitor at sites where the median annual radon concentration is around 1 or 2 Bq/m3 (as is likely to be the case for many ICOS stations)? Clearly, the ARMON V2 would be better suited as a calibration transfer standard for sites where median annual radon concentrations were above 5 Bq/m$^3$.

Lastly, following the "full uncertainty budget" for the ARMON V2 presented in this manuscript, the authors clarify in their response that the expected measurement uncertainty at the claimed detection limit of 0.13 Bq/m$^3$ is 60%, and at a concentration of 0.6 Bq/m$^3$ the uncertainty is 28%.

The plot below compares 30-minute output of the Saclay ICOS Station 100m radon detector with 30-minute AND hourly output of the ARMON V2, the subject of the current manuscript (this small data excerpt is from the field component of the intercomparison study currently in preparation).

[Figure]

As already mentioned, the absolute calibration of the Saclay 100m radon detector is still being finalised as part of the laboratory component of the independent intercomparison study mentioned above (and may be subject to change by a few %), and a full uncertainty budget of the Saclay 100m radon monitor (part of the same study), has yet to be published.

However, spectral analysis of the 30-minute concentrations of the Saclay 100m radon instrument is consistent with spectral behaviour of meteorological and trace gas observations measured at the same height; lending credence to fidelity of the reported 30-minute concentration variability (driven by various timescales of turbulence and atmospheric mixing) and indicating that all combined measurement uncertainties at these concentrations are likely to be small for this instrument. The spectral behaviour of the ARMON V2 begins to deviate from that of meteorological quantities at periods of atmospheric motion below 6 hours. This implies that most of the observed bias between the ARMON V2 and Saclay 100m detector results evident above are likely attributable to measurement uncertainty of the ARMON V2. This result is not unexpected when comparing instruments with detection efficiencies of around 0.006 cps/(Bq.m$^{-3}$) (ARMON V2) and 0.2 cps/(Bq.m$^{-3}$) (Saclay 100m detector).

As evident above, at radon concentrations generally between 2 to 5 Bq/m$^3$, the empirical measurement uncertainty of the ARMON V2 often exceeds 30%. From a counting perspective alone, this uncertainty will increase as concentrations decrease. Consequently, if the results of the "full uncertainty budget" presented in this study indicate an uncertainty of 28% at 0.6 Bq/m$^3$, I can only assume that some terms have either been underestimated or overlooked. I would appreciate any comment by the authors on how to reconcile these apparent theoretical and empirical discrepancies.

**Scott Chambers, ANSTO, Lucas Heights, NSW.**

---

## Author Comment (AC1)

**Authors' reply to CC1**: 'Comment on egusphere-2023-2680' by Scott Chambers, posted on 28 Nov 2023

In the present document the authors of the manuscript "Full characterization and calibration of a transfer standard monitor for atmospheric radon and thoron measurements", currently under review for publication in Atmospheric Measurement Techniques, want to answer point by point, to the comment posted by Scott Chambers. Authors answers are here reported in blue colour.

First of all, authors want to thank Scott Chambers to actively participate into the discussion phase of this manuscript. In the following lines clarifications are presented for each Scott's comment:

CC1: Abstract

-For better transparency (considering readers who may not be familiar with ISO 11929-4), it would be good in the abstract to provide the reader with context for the claimed ARMON v2 detection limit of 0.132 Bq m$^{-3}$ that would be directly comparable to other radon measurement systems. For example, some studies have shown the hourly measurement uncertainty of commercial AlphaGuard units at their nominal detection limit (of around 3 Bq m$^{-3}$) is 50 – 60%(in other cases the quoted uncertainty has been higher), and the radon concentration at which the 200 L ANSTO dual-flow-loop monitor has an hourly measurement error of 30% is around 0.14 – 0.16 Bq m$^{-3}$ (Chambers et al. 2022; doi:10.5194/adgeo-57-63-2022).

It is important to take in mind that the current manuscript wants to presents the characterization and calibration of a new version of the ARMON monitor. The comparison of the ARMON with others radon and/or radon progeny monitors has been performed during a different activity of the project traceRadon, its results are currently under analysis and they are going to be presented in a further manuscript where the uncertainty budget of the all monitors will be performed for atmospheric hourly radon concentration measured at a typical ICOS station.

However, to help the reader with the context of the work and to avoid faulty comparisons, we think it is important to underline that:

i. in absence of thoron, the background of the ARMON is zero, so any count detected of $^{218}$Po can be assigned to $^{222}$Rn, and therefore the decision threshold is zero. In fact, it could be declared that the detection limit of the ARMON is 1 count per hour (0.048 Bq m$^{-3}$), but the authors have opted to use the ISO-11929 definition. In addition, in the presented analysis of the full ARMON measurement uncertainty, all the uncertainties, those of type A from the counting and those of type B coming from the different variables that may affect the measure, are taken in consideration which may intrinsically depend or not from the instrument.

ii. in the paper cited by Scott Chambers, where the new 200L ANSTO monitor was presented, the full uncertainty budget of the radon concentration measured with this instrument was not performed and declared as in the present ARMON manuscript. Therefore, uncertainty values for both monitors cannot be compared without a complete evaluation of all the uncertainties of the ANSTO (e.g. uncertainty introduced by the background variability of the monitor, the flow sample variability, the deconvolution calculation application, the T/P/RH sensors, etc.).

iii. in the paper by Radulescu et al., 2022 ([https://doi.org/10.1016/j.nima.2021.165927](https://doi.org/10.1016/j.nima.2021.165927)) commercial radon monitors were compared only together with their statistical uncertainties.

Please take in mind again that the values reported are much bigger that 50-60% and, again, they do not represent the full uncertainty of the measurement.

- Based on the results of Figure 5a of this manuscript, the hourly ARMON v2 measurement uncertainty for $^{222}$Rn in a dry, $^{220}$Rn-free environment (i.e., best case scenario) at an ambient $^{222}$Rn activity concentration of around 0.6 Bq m$^{-3}$ is $\geq$ 30%. Guided by the shape of the curve in Figure 5a, the hourly measurement uncertainty at the claimed detection limit of 0.132 Bq m$^{-3}$ would likely exceed 100%. Stating the hourly measurement uncertainty along with the claimed detection limit in the abstract would be a better guide for the reader.

Figure 5a represents the total uncertainty obtained by the ARMON during real field atmospheric measurements and WITHOUT using the thoron delay volume. This means that this curve does not represent the ARMON best scenario because the thoron contribution, and its influence on the radon concentration uncertainty, may be low here but it is not zero. Actually, using the thoron decay volume will improve the results (Figure 5b). For example, the uncertainty at 0.6 Bq m$^{-3}$ is 28% (<30%), and 60% at the detection limit. We will clarify this point and add this value in the revised version of the manuscript.

-Furthermore, since the ARMON v2 is introduced here as being able to separately quantify radon ($^{222}$Rn) and thoron ($^{220}$Rn), it would be good to state in the abstract the expected detection limit and hourly measurement uncertainty both with, and without, the presence of $^{220}$Rn in the sampled airstream (assuming a representative $^{220}$Rn activity for the surface layer – such as the value quoted on Line 430).

Due to limitations in the number of words of the abstract it was not possible to include all results. However, we will try to rewrite it in the new version of the document to add this information too.

-Lastly, the suitable measurement range of the ARMON v2 is quoted to be 1 – 100 Bq m$^{-3}$, but the measurement uncertainty is given only for a concentration of > 5 Bq m$^{-3}$. Would it not make sense to quote the measurement uncertainty at 1 Bq m$^{-3}$? Or at least report this value also?

The full budget calculation was done for a typical inland atmospheric radon concentration value (5 Bq m$^{-3}$), however we will also add in the new version of the manuscript the range of variability of the radon uncertainty of the ARMON in the range between 1 and 100 Bq m$^{-3}$, which was the traceRadon project target.

CC1: Line 100

For completeness, the authors should also consider including the following paper in this summary: Wada, A., Murayama, S., Kondo, H.,Matsueda, H., Sawa, Y., and Tsuboi, K. (2010). Development of a compact and sensitive electrostatic Radon-222 measuring system for use in atmospheric observation. J. Meteorol. Soc. Jpn. 88, 123–134. doi: 10.2151/jmsj.2010-202.

Thank you Scott for the reference, we will add it at the corresponding line.

CC1: Section 2.4

Regarding the uncertainty and application of the STP correction for ARMON v2 measurements: according to Figure 1, the temperature measurements of the ARMON v2 are not made in the

measurement sphere, but a long way downstream of the sphere and some other instruments. The location of the pressure measurements is not indicated in the figure, it would be good to see where they are made. Given the separation between the sensors and measurement volume, and the fact that the temperature sensor is in a separate, ventilated compartment of the ARMON's transport case, can the authors give any indication of the expected additional uncertainty in the derived STP correction parameter? At the moment, it seems that only the instrument manufacturer uncertainty values for temperature and pressure are being considered.

First of all, we want to clarify that the ARMON does not have a pressure meter and the STP correction is performed using the pressure value and uncertainty from the atmospheric station pressure meter where the instrument is running, as the air inside the sphere is at atmospheric pressure because it is an open circuit. Regarding the temperature meter, in stationary measurements we do not think it may differ from the temperature inside the sphere, as it is located in the same box. This was confirmed in the past with an old version of the ARMON which had the sensor inside the detection volume (Grossi et al., 2012, https://doi.org/10.1016/j.radmeas.2011.11.006). In any case, the sensitivity study shows us that even if there was a big error in the temperature measurement (e.g. 3 ºC), the uncertainty added would be below 1%, very low compared to the elements that have a greater contribution to the uncertainty of the system. We will add a sentence in the modified version of the document to explain this fact.

General: Consider revising the text for grammatical accuracy.

Scott Chambers, ANSTO, Lucas Heights, NSW.

Thank you again and yes, in the revised version of the manuscript we will carry on a deep grammar revision too.

---

## Author Comment (AC2)

Dear Authors,

*Thank you for the response to my initial comments. I appreciate that the manuscript currently under review is not an intercomparison manuscript and I am not asking for it to become one. As the authors point out, a separate, independent characterization and intercomparison of the newARMON V2 with other radon monitors has recently been conducted, for which the results are still being finalized for publication. In itself, this brings into question the timing and utility of the present manuscript – given that its content may either detract substantially from the novelty of the planned independent intercomparison study or be difficult to retrospectively correct if later found to be incomplete.*

*Dear Scott,*

*Thank you again for your participation into this discussion.*

First of all, in this manuscript we present the instrument, the uncertainty budget, a theoretical approach on the efficiency against using a theoretical model, and the calibrations at two independent radon Chambers using different methodologies. The aim of it is introducing the instrument and its main characteristics to the scientific community as well as you did with your last paper where you presented the new ANSTO 200 L monitor (Chambers et al., 2022, doi:10.5194/adgeo-57-63-2022). We do not thing the future intercomparison paper, when and if it will come out, may focus on such instrument. For this reason, the content of the submitted manuscript will not detract anything to the future one. On the contrary, we think the manuscript in preparation may use and take benefit of the results showed in this work.

*The main point I was trying to make is that the transparency of reported findings is important to the research community. Every instrument has its strengths and weaknesses, and this information needs to be reported in an easily interpretable, objective manner for potential users to make an informed decision as to whether a particular instrument is fit for the purpose they intend to use it for.*

*Lines 31-32 claim the suitability of the ARMON V2 as a calibration transfer standard device, and Lines 50-54 imply that the main target instruments for this service would be part of the European ICOS network. Radon measurements as part of the ICOS network are typically made at heights of 100 – 200 m above ground level. Furthermore, some of these sites are at remote, coastal or mountaintop locations, where radon concentrations are usually well below reported values typical of the terrestrial boundary layer globally (the value quoted by the authors was 5 Bq/m³).*

*As an example, Cabauw is a key European ICOS site, located in a flat inland region. Based on 10 years of Cabauw radon observations at 200m agl, the 10th , 50th and 90th percentile radon concentrations for this site are 0.35, 1.1 and 3.6 Bq/m 3, respectively. Another relevant ICOS station is Saclay, in France. Based on measurements at this site in 2022 at 100m agl the 10th , 50th and 90th percentile radon concentrations were 1, 2.5 and 6.1 Bq/m³, respectively. If the uncertainty of the ARMON V2 is only less than 10% for radon concentrations above 5 Bq/m³, would the authors be able to comment on the implications for transferring an SI traceable calibration to an operating radon monitor at sites where the median annual radon concentration is around 1 or 2 Bq/m3 (as is likely to be the case for many ICOS stations)? Clearly, the ARMON V2 would be better suited as a calibration transfer standard for sites where median annual radon concentrations were above 5 Bq/m³.*

Please take in mind that the idea of this manuscript is to show ARMON monitor and its capabilities. We are showing the full uncertainty for a whole range, taking in consideration all variables that may affect the measure, and not just the Standard deviation or the short-term variability as it is usually done. We state that it could be used as a Transfer Standard because, as shown within the manuscript, i) its calibration factor does not depend on the activity calibration range; ii) the monitor can be displace to atmospheric stations; iii) the monitor has not background when used with a thoron delay volume; iv)it can allow the spectra analysis of all radon radionuclides and, among others, it accomplishes with the requirement of the traceRadon project objective (WP1).

*Lastly, following the "full uncertainty budget" for the ARMON V2 presented in this manuscript, the authors clarify in their response that the expected measurement uncertainty at the claimed detection limit of 0.13 Bq/m 3 is 60%, and at a concentration of 0.6 Bq/m 3 the uncertainty is 28%.*

*The plot below compares 30-minute output of the Saclay ICOS Station 100m radon detector with 30-minute AND hourly output of the ARMON V2, the subject of the current manuscript (this small data excerpt is from the field component of the intercomparison study currently in preparation).*

*As already mentioned, the absolute calibration of the Saclay 100m radon detector is still being finalised as part of the laboratory component of the independent intercomparison study mentioned above (and may be subject to change by a few %), and a full uncertainty budget of the Saclay 100m radon monitor (part of the same study), has yet to be published.*

*However, spectral analysis of the 30-minute concentrations of the Saclay 100m radon instrument is consistent with spectral behavior of meteorological and trace gas observations measured at the same height; lending credence to fidelity of the reported 30-minute concentration variability (driven by various timescales of turbulence and atmospheric mixing) and indicating that all combined measurement uncertainties at these concentrations are likely to be small for this instrument. The spectral behaviour of the ARMON V2 begins to deviate from that of meteorological quantities at periods of atmospheric motion below 6 hours. This implies that most of the observed bias between the ARMON V2 and Saclay 100m detector results evident above are likely attributable to measurement uncertainty of the ARMON V2. This result is not unexpected when comparing instruments with detection efficiencies of around 0.006 cps/(Bq.m$^{-3}$) (ARMON V2) and 0.2 cps/(Bq.m$^{-3}$) (Saclay 100m detector).*

*As evident above, at radon concentrations generally between 2 to 5 Bq/m$^3$, the empirical measurement uncertainty of the ARMON V2 often exceeds 30%. From a counting perspective alone, this uncertainty will increase as concentrations decrease. Consequently, if the results of the "full uncertainty budget" presented in this study indicate an uncertainty of 28% at 0.6 Bq/m$^3$, I can only assume that some terms have either been underestimated or overlooked. I would appreciate any comment by the authors on how to reconcile these apparent theoretical and empirical discrepancies.*

Concerning the data you are showing, we have some remarks to show them publicly because:

- The data of the ARMON that you have shown from the intercomparison at Saclay carried out within the traceRadon project were flagged as "possible leak" and it should not be used for the analysis. This is the reason because we think that not full analyzed data may not have been used and mainly without previously discussing it with the other authors.
- You are also mainly showing half-hour values (hourly values in light grey are really difficult to distinguish), while the uncertainty we are giving is always in hourly values which is the time resolution used at ICOS atmospheric stations.
- It is already known that the deconvolution algorithms used by the ANSTO monitor could produce artifacts such as the smoothing the data when there are quick changes in concentrations, so we must take care in accept as real ones the activities given by the ANSTO although your study on atmospheric mixing. As you already know when an intercomparison is carried out between two instruments no one presents the reference value.

Finally, in the calibrations at PTB, we have found that the empirical uncertainties at low concentrations (the deviations around an estimated value in the calibration chamber) are in total agreement with the ones estimated, so there is no discrepancy at all. Stefan Röttger is currently preparing another manuscript on the full calibration procedures.

Best Regards

---

## Author Comment (AC3)

**Answer to reviewer 1**

*The manuscript describes an instrument sensitive to radon and thoron in concentrations frequently found in near-surface air above continents. Characterisation and calibration of the instrument were thorough, though only for radon and not for thoron (lines 114-115). Hence, the manuscript title requires modification. The instrument can help to make sure that radon concentrations measured at different stations within a monitoring network are real and unaffected by differences between instruments' performance, their individual calibration, or differences in data processing.*

- First of all, authors want to thank the reviewer for his/her time. We have now removed the reference to thoron within the title and modified the abstract in relation to it. Although the instrument is capable to measure thoron, is true that it was not calibrated and characterized for this gas measurements.

*There is little I can add to the earlier discussion between Scott Chambers and the Authors. The main additional issue I would like to raise is the long-term stability of the instrument's calibration. Is the instrument assumed to maintain its current calibration throughout its lifetime, even when travelling frequently and extensively? Or, is regular re-calibration foreseen? If so, how often? Lines 496 to 499 hint at a calibration unit for "...very short calibration or recalibration [...] under field conditions..." The next sentence states this issue "...will be the object of a future paper." There are two further papers announced, one on a field intercomparison with an ANSTO detector (lines 499 to 501) and another one on the "full calibration procedures" (last line in AC2). Is it really necessary to distribute the outcome of this enterprise among four (!) papers? From my point of view, the mobile calibration unit definitely has to be included in the present manuscript, as should be more details about the air dryer and the thoron delay volume that will go with the instrument once it will be 'on the road' as a travelling standard instrument. Please add these items also to Figures 1 and A2. In addition, please show in the schematic diagram of Fig. 1 the position of the air pump.*

- In regard to the long-term stability of the instrument's calibration, a previous ARMON version used at the Spanish station of Gredos and Iruelas (Grossi et al., 2018) and now running at the Barcelona station, was calibrated after several years of being in the field at the INTE radon chamber and after travelling over 800 km by car. In this paper the ARMON v2 calibration factor obtained at the INTE chamber (Barcelona, Spain) was compared with the one obtained at the PTB (Braunschweig, Germany) where the instrument arrived 18 months late after travelling by car for the traceRadon project campaigns from Barcelona to the PTB, then to Saclay, France and back again to the PTB. This point has been now clarified within the manuscript with the following text (section 3.4):

'Results of the calibration at PTB, done 18 months after the calibration at INTE, also confirm that the calibration of the instrument is stable over the time, as it was already appreciated in the older version of the monitor (Grossi et al., 2012, 2018; 2020; Vargas et al., 2015). However, in a mark of calibration procedures of radon measurement network it is suggested to perform periodical stability checks of the efficiency of the diferent radon and radon progeny instruments running at the diferent stations'.

- In regard to the number of papers where the results of the ARMON calibration and characterization, of the calibration procedure of radon and radon progeny monitors (not only the ARMON) running at atmospheric stations and of the intercomparison of different radon and radon progeny monitors will be presented, the scientific community of the traceRadon project already decided to create a Special Issue in the journal of Atmospheric Measurement Techniques where the different outputs from the different authors and over the years will be published under the same umbrella. This division will help the readers to easily follow the different research developments. More Info here: https://amt.copernicus.org/articles/special_issue1257.html
- Finally, in the present manuscript as suggested by the reviewer the drying unit and the pump were added to Figure 1 and A2. The calibration procedure of the ARMON will be different depending if the station needs or not a drying unit for this instrument. Actually, we have observed that there are stations (as for example the ICOS station of Saclay, France) that can deliver dry sample air, so no drying unit is needed, while in others a drying unit may be used. Anyway, the drying unit with a delay unit is now presented in Appendix C.

*At the end of AC1 you state that "... air inside the sphere is at atmospheric pressure because it is an open circuit." This presumption cannot be correct. If air pressure inside the sphere would be exactly the same as outside, there would be no air flowing through the sphere. Yet, the sphere is continuously flushed with 2 L/min. Upstream of the sphere a filter, downstream a flow meter (Fig. 1). Both restrict flow in addition to the tubing connecting the sphere with the outside. It may not be necessary to continuously monitor pressure inside the sphere, but I would suggest to determine once the pressure difference between inside and outside the sphere and add the offset to the pressure reading from the atmospheric station. Even if the correction is small, it should be included because not doing so introduces a perhaps small but systematic error.*

- Thank the reviewer for this suggestion. We have done the calculation assuming a high pressure difference between inside and outside the ARMON detection volume and the uncertainty is really small. The error induced in the radon concentration due to errors of pressure or temperature has been now added in section 3.3:

*'As for the STP correction, the values of T and P uncertainties have been taken from the sensor uncertainties. A higher uncertainty could be due to the distnace between the the sensors positon and the detection volume of the instrument. However, calculus show that these uncertantiies will be negligible. Let the Reader consider that an increase of the temperature uncertainty of 2 degrees will suppose an increase in the uncertainty of $1.4 \cdot 10^{-3}$ Bq m$^{-3}$, and an increase of 5 hPa in the uncertainty of Pressure will only increase total uncertainty by $4 \cdot 10^{-3}$ Bq m$^{-3}$'.*

**3.) Perhaps tell the reader already in line 139 that the detection volume is 20 L.**

- The correction has been applied as suggested.

**4.) Line 445: The humidity was < 150 ppm, so why Eq. 2 and not Eq. 13?**

- We have now compared both calibrations (INTE and PTB) using the exponential fit (Eq. 13).

---

## Author Comment (AC4)

**Answer to Reviewer 2:**

*The authors present a detailed description of the new version of the ARMON detector, including its metrological characterization. In particular, I appreciate the carefully done uncertainty budget.*

*The ms. is well structured and written, motivation and conclusions are clear.*

*Att. the commented ms. pdf. Most comments are trivial linguistic suggestions which the authors are free to accept or not. One perhaps more serious comment pertains to the simulation technique in sec. 3.1 / fig. 2b.*

*Overall a very interesting paper!*

First of all, authors want to thank a lot the reviewer for his/her positive feedback. All comments and linguistic suggestions have been now included within the revised version manuscript. We have also modified the figure 3 and now $\varepsilon_0$ and $\varepsilon_0'$ are marked in order to help readers.

As for your question about the simulation:

*Has the simulation performed in 2D or 3D? This makes a difference. If it was in 3D, then the picture is the projection of the 3D particle locations in the sphere onto a 2D disk, which would explain that density appears lower near the border. However - just by feeling! - I would expect higher apparent particle density near the centre.*

The simulation was done in a 3D sphere with particles homogenously distributed within the all volume. In figures 2b and 2c, we have just represented "z" vs "x" (for all y's), and therefore there are more particles in the middle.

The figures below represent two "plotting versions" of the distribution of particles inside the sphere. On the left we have plotted "z" vs "x" as it appears in the manuscript, and on the right, we have represented the radius (distance to "z" axis, with negative values when x<0) instead of "x". We hope this clarification will help the reviewer.